# The spatiotemporal evolution of atmospheric boundary layers over a thermally heterogeneous landscape

**Mary Rose Mangan**[1,⚲], **Jordi Vilà-Guerau de Arellano**[1], **Bart J. H. van Stratum**[1], **Marie Lothon**[2], **Guylaine Canut-Rocafort**[3], **and Oscar K. Hartogensis**[1]

[1]Meteorology and Air Quality, Wageningen University & Research, Droevendaalsesteeg 4, 6708 PB, Wageningen, the Netherlands
[2]Laboratoire d'Aérologie, Université de Toulouse, CNRS, UPS, 14 avenue Edouard Belin 31400, Toulouse, France
[3]Centre National de Recherches Météorologiques (CNRM)/Météo-France, 42 ave. G. Coriolis, 31057, Toulouse, France
[⚲]*Invited contribution by Mary Rose Mangan, recipient of the EGU Atmospheric Sciences Outstanding Student and PhD candidate Presentation Award 2022.*

**Correspondence:** Mary Rose Mangan (maryrose.mangan@wur.nl)

**Abstract.** We study the diurnal variability in the atmospheric boundary layer (ABL) across spatial scales (between $\sim 100$ m and $\sim 10$ km) of irrigation-driven surface heterogeneity in the semi-arid landscape of the 2021 Land surface Interactions with the Atmosphere over the Iberian Semi-arid Environment (LIAISE) experiment on the northeastern Iberian Peninsula. We combine observational analysis with explicit simulation of the ABL using observationally driven large-eddy simulation (LES) to better understand the physical mechanisms controlling ABL dynamics in heterogeneous regions. Our choice of spatial scales represents current and future single grid cells of global models, demonstrating how the sources and magnitude of subgrid-scale heterogeneity vary with model resolution.

There is an observed positive buoyancy flux over the irrigated fields driven primarily by moisture fluxes, whereas, over the non-irrigated fields, there is a classical buoyancy profile driven by the surface sensible heat flux. The surface heterogeneity is felt most strongly near the surface; however, at approximately 1000 m above the surface, there appears to be a blending zone of mean scalars (i.e., potential temperature and specific humidity), indicating that the heterogeneity mixes into a new mean state of the atmosphere. There is a stable internal boundary layer (IBL; as defined as the first stable layer in individual radiosonde potential temperature profiles) up to approximately 500 m over the irrigated area. Taking advantage of the spatiotemporal extent of LES results, we perform spectral analyses to find that the ABL height had an integral length scale of $\sim 800$ m matching that of the imposed surface fluxes. Between the irrigated and non-irrigated areas, there is an adjustment of the ABL as it crosses the boundary up to 500 m upwind of the boundary. We observe a variable-dependent blending zone between scales in the middle of the ABL, but it is limited by the entrainment zone effectively introducing another source of heterogeneity driven by upper-atmosphere conditions.

# 1 Introduction

Surface heterogeneity impacts the development of the atmospheric boundary layer (ABL) in a number of ways depending on the atmospheric stability and the strength, size, and orientation of the surface heterogeneity (Bou-Zeid et al., 2020; Brunsell et al., 2011; Hechtel et al., 1990; Huang and Margulis, 2009; Patton et al., 2005; Shen and Leclerc, 1995; van Heerwaarden et al., 2014). Generally, the process-based impacts of surface heterogeneity can be reduced to two classes: (1) the formation of an internal boundary layer (IBL) and (2) the formation of secondary circulations modulated by the meso- and synoptic scales. In this study, we use a combination of observational data from the Land surface Interactions with the Atmosphere over the Iberian Semi-arid Environment (LIAISE) field experiment and a large-eddy simulation (LES) inspired by the results of the observational study to investigate the impacts of unstructured, realistic surface heterogeneity on the development of the ABL. In the LIAISE domain, locally applied irrigation creates a thermal surface heterogeneity. We hypothesize that, based on the scale of heterogeneity, the regional LIAISE boundary layer is a composite of a representative ABL from the wet and dry patches (Mangan et al., 2023a). We are motivated to answer the following general research question:

*What physical processes dominate the spatiotemporal evolution of the ABL in a case with realistic surface heterogeneity?*

The vertical impact of the surface heterogeneity is largely a function of local stability. Under near-neutral conditions, an internal boundary layer (IBL) can form at the boundary between two patches (Garratt, 1990). Internal boundary layers form when the background wind flows across the boundary and at the boundary of the heterogeneity, and they adjust downwind as they equilibrate with the surface. Typically, IBLs are identified by two mixed layers in scalar profiles or by vertical flux divergence near the surface (Mahrt, 2000). Mahrt (2000) stresses that both mesoscale and microscale IBLs can be formed depending on the patchiness of the land surface and the length scale of the heterogeneity.

Under low wind and low convective conditions, secondary circulations can develop (Avissar and Schmidt, 1998; Liu et al., 2011; Maronga et al., 2013; Ouwersloot et al., 2011; Patton et al., 2005; Raasch and Harbusch, 2001; Shen and Leclerc, 1995; van Heerwaarden and Vilà-Guerau de Arellano, 2008). Avissar and Schmidt (1998) and Raasch and Harbusch (2001) found that background winds between 5 and $7\,\mathrm{m\,s^{-1}}$ eliminate the impact of a secondary circulation, but the effect of the background wind depends on the orientation of the wind with respect to the boundary. If the background wind is not favorable for the formation of secondary circulations (for example, it does not flow perpendicular to the boundary), then wind speeds as low as $2.5\,\mathrm{m\,s^{-1}}$ are high enough to destroy their formation. In addition to the background wind and stability, secondary circulations also depend on the scale of heterogeneity with respect to the ABL depth and ABL turbulence scales. The ABL is most impacted when the scale of the heterogeneity is on the same order of magnitude as the ABL depth (Patton et al., 2005; Raasch and Harbusch, 2001; Shen and Leclerc, 1995; van Heerwaarden et al., 2014).

Because the impacts of surface heterogeneity on the ABL can occur on smaller spatial scales than the ones explicitly resolved by regional and global weather models, subgrid cell heterogeneity is poorly represented. These impacts are represented either by aggregating the land surface properties to create a composite surface that provides one flux to the atmosphere (e.g., parameter aggregation) or with a tiled approach, where non-interacting tiles of different land surfaces provide a flux to the lowest model level that blends in the atmosphere as in ECMWF's ERA5 reanalysis model (Bou-Zeid et al., 2020). In the latter method, the blending height is a useful concept for describing how the impacts of surface heterogeneity are projected in the ABL. Unlike the physical processes of the IBLs and secondary circulations, blending height is a concept that arises from the practical need that our theory and numerical models necessitate a homogeneous surface. For example, in global-scale numerical models, impacts of subgrid-scale surface heterogeneity blend in a composite ABL below the lowest grid cell of the model (Bou-Zeid et al., 2020). In this study, we define blending height in the same way as Mahrt (2000) as a scaling depth that describes the decreasing influence of surface heterogeneity on the atmosphere with height. It is not necessarily a physical level where heterogeneity is no longer discernible; instead, it is a threshold where the heterogeneity becomes negligible compared to a homogeneous case from the perspective of researchers.

Previous studies that have focused on the array of ways that surface heterogeneity can impact the ABL have either been (1) observational studies which capture the realistic surface heterogeneities' impact on the ABL on a limited spatial scale or (2) idealized large-eddy simulation (LES) and direct numerical simulation (DNS) studies which create scaling theories to study how the heterogeneity can impact the ABL. Observational studies capture the realistic surfaces and heterogeneities, but measurements are typically unable to capture the full extent of the processes that govern the ABL response to the surface heterogeneity. Some examples of observational studies in heterogeneous areas include LITFASS-2003 (Beyrich and Mengelkamp, 2006), CHEESEHEAD (Butterworth et al., 2021), and GRAINEX (Rappin et al., 2021). While the observational studies focus on a relatively small spatial scale, the idealized studies focus on idealized LES to write "textbook" cases of how surface heterogeneity impacts the ABL. There have been a few studies using realistic or real surface in LES, including Hechtel et al. (1990); Huang and Margulis (2009) and Maronga et al.

(2013). Because the impact of the surface heterogeneity on the ABL depends on the scale and strength of heterogeneity and the stability, it is unknown how and when these features of surface heterogeneity impact the ABL dynamics over the course of a realistic, unstructured heterogeneity on a convective day.

The purpose of this study is to evaluate the impacts of surface heterogeneity on the ABL across three scales over a representative LIAISE day, both from a data-driven approach with the comprehensive LIAISE field campaign and from a modeling-driven approach with a high-resolution LES. In this way, we can evaluate both the physical nature of how the surface heterogeneity impacts the ABL and the potential impacts of how resolved turbulence blends the heterogeneity with height in the atmosphere. Our approach combines observational data with an LES experiment. Firstly, we explore the LIAISE ABL using observations. We study boundary layer development spatially and temporally by combining a network of surface energy balance stations, radiosondes, and aircraft data (Sect. 2). Based on the results from Sect. 2, we define more explicit sub-research questions to be explored with LES (Sect. 3). In Sect. 4, we run an LES experiment inspired by the observations of the LIAISE composite day to study how the observed ABLs form across different spatial scales. For the LES, we prescribe the land surface with observations so that we can capture the realistic, unstructured surface heterogeneity that was observed during the LIAISE experiment. Finally, in the Discussion (Sect. 5), we bring model and data results together to answer the question of how the development of the ABL differs across spatial scales of heterogeneity in the LIAISE experiment. In particular, using the LES, we investigate the characteristic length scales of heterogeneity that propagate into the ABL, and we discuss how the scales of surface heterogeneity blend in the ABL.

## 2 The LIAISE experiment

The LIAISE field experiment took place between April and October 2021 in the Ebro River Valley in Catalonia on the northeastern Iberian Peninsula, with an intensive observation period (IOP) occurring in July 2021 (Boone et al., 2021, 2025; Mangan et al., 2023a). In particular, we focus our study on a period of 3 d in the middle of the IOP, 20–22 July 2021. The extent of the LIAISE experiment was characterized by a thermal surface heterogeneity due to locally applied irrigation in agricultural fields. Although the atmospheric conditions observed in the LIAISE experiment are controlled in part by the surface heterogeneity, there is strong coupling between the synoptic-scale processes and the regional-scale land surface that complicate the dynamics of the ABL. For that reason, we selected these days with relatively weak synoptic forcing.

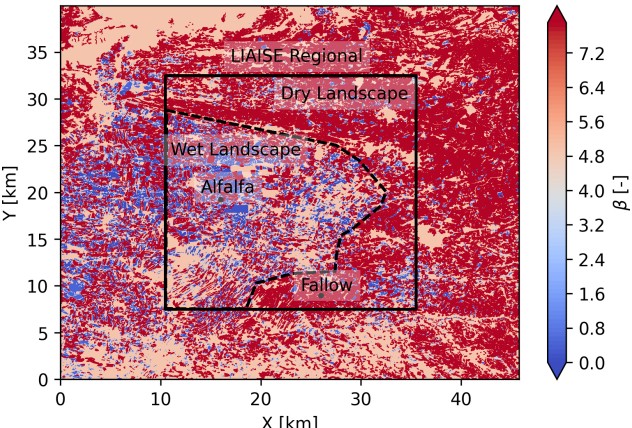

**Figure 1.** The observed surface Bowen ratio at 12:00 UTC in the domain of the LIAISE LES. The spatial scales are indicated on the map with "LIAISE Regional" (bounded box), "Wet Landscape", "Dry Landscape" (dashed line), "Alfalfa Local", and "Fallow Local" (points).

During the 3 d period, there was a thermal low building to the west of the LIAISE domain, which was influential in controlling the surface winds and the boundary layer development. Figure 1 shows the observed Bowen ratio ($\beta \equiv H/LE$) for the extent of the LIAISE experiment at 12:00 UTC averaged over 20–22 July 2021 (Mangan et al., 2023a). In the irrigated area (western side of the domain), $\beta$ is as low as 0.01; in the rainfed area (eastern side of the domain), $\beta$ is as high as 20. The difference in crop type and irrigation causes a strong heterogeneity with a length scale on the order of 10 times the ABL height, so one would expect a strong influence of this heterogeneity on the ABL.

### 2.1 Relevant spatial scales

There are a number of relevant spatial scales of surface heterogeneity that influence the atmospheric flow in the LIAISE domain. The largest scale of influence is the synoptic scale, which is important for controlling the mean wind and subsidence around our study area. The selected days of study, 20–22 July 2021, are characterized by the development of a thermal low in the northern center of the Iberian Peninsula to the west of the study area (Hoinka and Castro, 2003). In addition to the thermal low, in the late afternoons, the LIAISE area is also influenced by a sea breeze from the Mediterranean Sea (Jiménez et al., 2023; Lunel et al., 2024b). At the end of the afternoon, a cool and moist easterly sea breeze reaches the LIAISE region, suppressing the ABL growth. Depending on the location of the thermal low, it can enhance or diminish the strength of the sea breeze.

Although these synoptic- and mesoscale features define the environment, the ABL is also influenced by smaller spatial scales. We focus our study on the three scales of spatial heterogeneities as defined by Mangan et al. (2023a, b):

the LIAISE regional scale, the wet and dry landscape scales, and the alfalfa and fallow local scales (indicated in Fig. 1). The regional scale ($\sim 10$ km; $\sim 10$ ABL height [$z_i$]), which was the largest defined, consists of both the irrigated and the non-irrigated areas. It is the area of a single ERA5 grid cell. Within the regional scale, there are two landscape scales ($\sim 1$ km; $\sim 1 z_i$): the wet and dry landscape scales. Each landscape scale falls within the irrigated and non-irrigated areas, respectively. They are characterized by the heterogeneity that arises between fields due to agriculture type and irrigation schedule. The separation between the wet and the dry landscape scales is shown with a dashed line in Fig. 1. The smallest scale defined is the local or field scale consisting of homogeneous individual fields ($\sim 100$ m; $\sim 0.1 z_i$), the irrigated alfalfa fields and rainfed fallow fields where much of the measurements were taken. These scales are shown by points in Fig. 1.

## 2.2   LIAISE data

During the intensive observation period in July 2021, there were high spatial and temporal resolutions of measurements of the land surface and the ABL. The surface and surface layer observations came from a network of surface energy budget (SEB) stations in nine of the predominant crop types in the LIAISE domain. The fluxes from each SEB station were applied to a high-resolution crop-cover map to create spatial estimates of fluxes in the LIAISE regional domain. In addition to SEB stations, Catalonia has a dense network of automated weather stations operated by Servei Meteorològic de Catalunya (https://www.meteo.cat/observacions/xema, last access: 2 July 2025). The stations around the LIAISE domain were used to estimate advection of temperature, moisture, and wind using 10 m wind and 2 m temperature and humidity measurements. Both the flux maps and the advection calculation are described in more detail in Mangan et al. (2023a). The surface layer was probed by two 50 m towers: one located in an irrigated alfalfa field and one in a non-irrigated fallow field. These 50 m towers included sensible heat flux measurements at 3, 10, 25, and 50 m and latent heat flux at 3 and 50 m. Furthermore, the 50 m towers were equipped to measure wind, temperature, and humidity profiles.

In addition to the surface and surface layer observations, there were a number of measurements of the ABL located in both the alfalfa fields and the fallow fields. During the IOP, hourly radiosondes were launched at each site between 06:00 and 18:00 UTC. There was also a tethersonde in the alfalfa field which was repositioned vertically approximately hourly during the daytime. Typically, the tethersondes measured turbulent fluxes, including those of moisture and buoyancy between 100 and 500 m a.g.l. at a time resolution of approximately 30 min. Finally, the SAFIRE aircraft (as described by Brilouet et al., 2021) measured turbulent fluxes of buoyancy, moisture, and momentum in the ABL once per day during the IOP. The measurement strategy included flying "legs" over each of the irrigated and non-irrigated areas at heights of 600–2000 m a.g.l. and a transect across the wet–dry transition between the alfalfa and fallow fields at approximately 1500 m a.g.l. On all days, the aircraft flew between 12:00–16:00 UTC.

We have chosen to use a "composite day" in this experiment to capture the inter-day variability in the atmosphere during the LIAISE experiment. The composite day is composed of 3 d, 20–22 July 2021, which are characterized by the thermal low that developed in the northern central area of the Iberian Peninsula. During these days, synoptic conditions were similar and relatively weak. Over each of the days, there was a thermal low that developed in the Ebro River Valley, and, in the late afternoons, a sea breeze bringing relatively cool and moist air from the Mediterranean Sea arrived in the study domain. Before the sea breeze, at the surface winds were primarily weak ($\approx 0$–$3$ m s$^{-1}$) from the west/northwest. Wind speeds reached up to 5 m s$^{-1}$ in the middle of the ABL. Furthermore, there was little daily variably in surface fluxes at each of the SEB sites over this period.

We create the composite day by averaging the available data per 30 min period across all days. We do this both for surface fluxes and for the atmospheric fields from the radiosondes, including wind speed, specific humidity, and potential temperature (mean values and daily variability shown in Fig. 2). The benefits of using a composite day for analysis include (1) gap-filling missing data, (2) reducing the spikes in the data, and (3) creating an atmospheric situation that is typical for the area during a thermal low day. Note that all processing (e.g., computing turbulent fluxes) and normalization were done on individual observations before averaging to create the composite day. This way, we create a situation that is realistic for the LIAISE experiment but is not a real/case study simulation. By using a composite-day approach, we can focus on the most persistent and important processes that control land–atmosphere interactions across the spatial scales.

## 2.3   Observed ABL from the LIAISE experiment

Boundary layer observations from the composite day show the diurnal evolution of the boundary layer potential temperature and specific humidity profiles in the wet and dry landscapes of the LIAISE campaign (Fig. 2). In the morning, between 08:00 and 10:00 UTC, we observe stable profiles in both landscapes. Temperature differences between the irrigated and non-irrigated profiles are small above 1 km; however, even by 10:00 UTC, the surface layer ($0$ m $< z <\sim 100$ m) in the wet landscape is wetter than that of the dry landscape. At 12:00 UTC, a well-mixed convective boundary layer begins to form. In the wet area, there are two distinct, relatively well-mixed layers in the temperature profile: one from the surface to 500 m and one between 500–1000 m. As a result of averaging to a LIAISE composite day, there

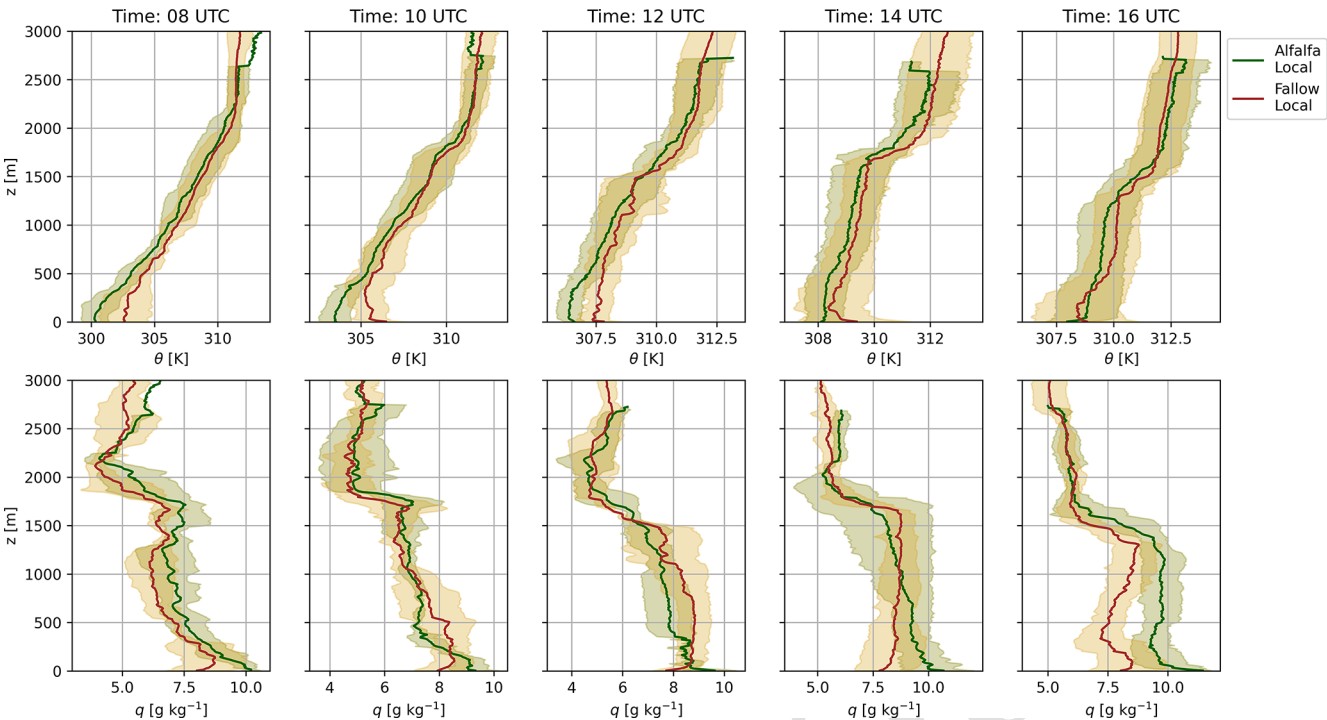

**Figure 2.** The radiosondes launched at the alfalfa field (green) and the fallow field (gold) averaged over 20–22 July 2021 at 08:00, 10:00, 12:00, 14:00, and 16:00 UTC. The shading is the daily variability across the days that comprise the LIAISE composite day. The top panels are the potential temperature profiles, and the bottom panels are the specific humidity profiles.

is a stable layer near the top of the ABL in both the alfalfa fields and fallow fields. This layering is not apparent in the humidity profile. The layering in the wet landscape radiosonde data continues in the afternoon at 14:00 UTC, while the dry landscape radiosonde data show a more traditional single well-mixed profile. Moreover, there is a strong signal of either entrainment or advection in the humidity profiles as the specific humidity in the wet area decreases with height towards the top of the ABL. This could be connected to the presence of a secondary circulation which could increase entrainment as shown by van Heerwaarden and Vilà-Guerau de Arellano (2008). In the mid-afternoon, the top of the ABL is still cooler in the wet landscape than the dry landscape. This is likely due to advection, but, from observations alone, we cannot quantify the advection between the wet and dry landscapes. By the end of the afternoon, at 16:00 UTC, the sea breeze arrives at the LIAISE region. The near-surface atmosphere cools and moistens in both locations, but the remainder of the profile is well mixed.

The radiosondes show the influence of the observation footprint. Near the surface, the profiles show more extreme gradients: unstable layers near the surface in the dry area and moist, internal boundary layers in the wet area. From observations, the IBL height ($z_{IBL}$) was determined by the first stable layer in the potential temperature profile above the surface (Appendix A). However, the profiles show similar mean values near the top of the ABL (Appendix A). This suggests

that there is some sort of "blending height" for the conserved variables of potential temperature and specific humidity between the wet and dry areas within the ABL. Previous LIAISE studies hypothesized this "funnel" type of ABL where, near the surface, observations are linked to local fields and, near the top of the ABL, observations represent a composite of both the wet and dry areas (Mangan et al., 2023a).

In addition to scalar profiles of the ABL from radiosondes, there were a number of different platforms to measure turbulent fluxes within and above the ABL. Figure 3 shows profiles of turbulent fluxes of heat, water vapor, buoyancy, and turbulent kinetic energy (TKE) in both the wet and dry areas. The profiles are normalized by the ABL height ($z_i$) derived from the parcel method (using a d$T = 1.25$ K) using the potential temperatures from the radiosondes (Kaimal and Finnigan, 1994). The observations are averaged both over the 3 composite days and the over hours 13:00–16:00 UTC to create the composite profile. Although averaging the observations over both the afternoons and with time could introduce errors, we assume that, because each observation is normalized by observed $z_i$ for each time individually, the scaling is reasonable for a convective ABL, although the surface fluxes may differ. The standard deviation, which is shown with the error bars in Fig. 3, provides an indication of the diurnal and inter-day variability in the observations. The mean ABL heights for these times are shown in Appendix A.

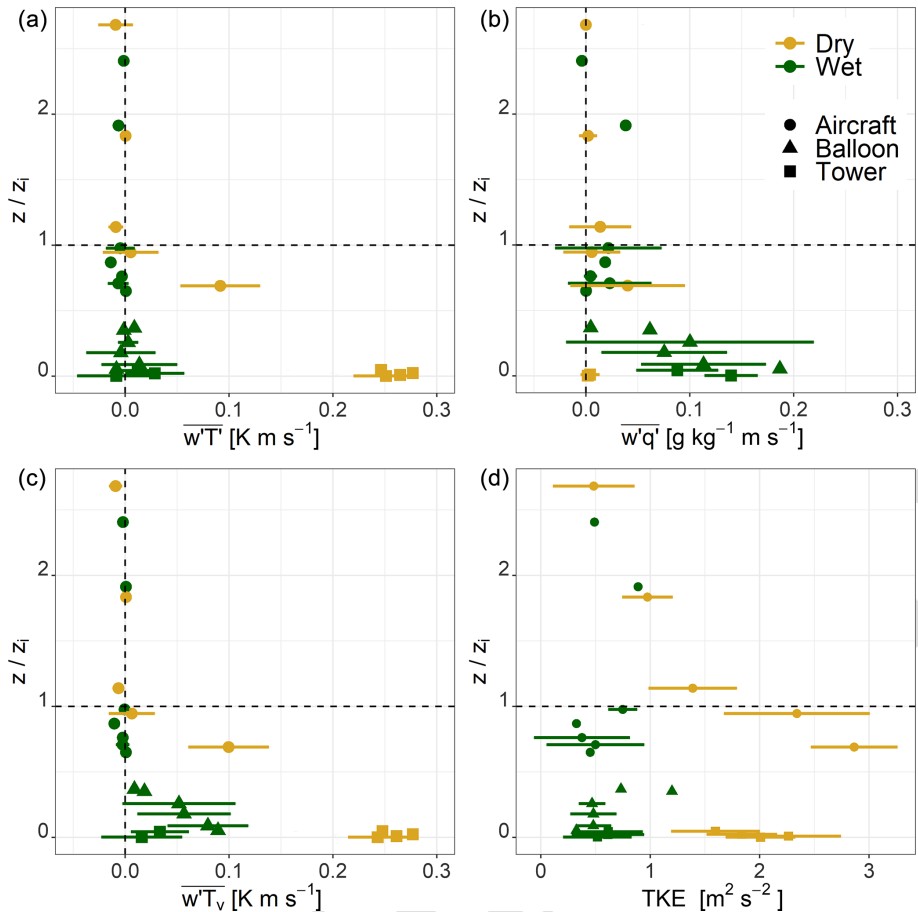

**Figure 3.** Composite flux profiles constructed at the alfalfa site (green) and the fallow site (gold). The vertical height is normalized by the boundary layer height ($z_i$) from the radiosondes. The profiles are constructed using data from 20–22 July 2021 from 13:00–16:00 UTC. **(a)** The kinematic heat flux, **(b)** the moisture flux, **(c)** the virtual potential temperature (e.g., buoyancy) flux, and **(d)** the turbulent kinetic energy. The error bars arise from the averaging over the days 20–22 July 2021 and the hours 13:00–16:00 UTC and from binning the observations by $z/z_i$.

In the dry landscape, the turbulent fluxes follow traditional textbook profiles of the convective boundary layer (Fig. 3). At the surface, there is the strongest sensible heat flux, and it decreases nearly linearly with height. This is consistent with the quasi-steady-state approximation for potential temperature in the mixed layer. Near the top of the ABL, the heat flux is near zero. Above the ABL, the heat flux is small. In the dry area, there is a virtually no moisture flux near the surface, but this increases near the top of the boundary layer because of entrainment. The combination of the negative heat flux and positive moisture flux near the top of the boundary layer is indicative of entrainment from the free troposphere. In this landscape, the buoyancy flux is dominated by the signature of the heat flux and follows the prototypical convective ABL flux patterns (Lenschow and Stankov, 1986; Stull and Driedonks, 1987). These turbulent fluxes of scalars are driven by TKE. In the dry landscape, there is relatively strong TKE throughout the column of the ABL. Near the surface, the TKE is driven by the horizontal velocity components ($u'^2$

and $v'^2$); however, near the top of the ABL, where convective thermals are stronger, the turbulence is more component-wise isotropic. At the top of the ABL in the dry landscape, $u'^2 \approx v'^2 \approx w'^2 \approx 0.5\,\mathrm{m^2\,s^{-2}}$ and is compared to in the surface layer, where $u'^2 \approx v'^2 \approx 0.5$ and $w'^2 \approx 0.1\,\mathrm{m^2\,s^{-2}}$.

Conversely, in the wet landscape, the turbulent heat flux is near zero from the surface through the entire ABL (Fig. 3). Near the surface, the error bars are large and show that there is uncertainty in the sign of the heat flux. Although the heat flux is small, the moisture flux in the wet area is high ($\sim 0.1\,\mathrm{g\,kg^{-1}\,m\,s^{-1}}$) from the surface through the lower half of the boundary layer. Above this layer, the moisture fluxes are near zero before showing a slight increase near the top of the ABL. Unlike in the dry area, in the wet area, the buoyancy flux has a large contribution from the moisture flux. In the bottom half of ABL, the moisture flux accounts for 20 %–50 % of the total buoyancy flux. At the surface, where heat flux is negative, there is a positive buoyancy flux because of the contribution of moisture. The buoyancy

flux peaks at about 25 % of the boundary layer height. From there, it decreases to near zero by 50 % of the boundary layer height. The scalar fluxes are small in part because the TKE is smaller than the dry area and remains constant through the ABL. Near the surface, the TKE is driven by shear, like in the dry area, but the velocity variances are smaller than in the dry area. In the middle of the ABL, the majority of the TKE difference between landscapes arises from the variance in vertical velocity. The buoyant thermals in the wet landscape are weaker than in the dry landscape, so, overall, the mixing of turbulence is weaker in the wet area than in the dry area. Both the TKE and buoyancy profiles suggest that there is an IBL from the surface up until $z/z_i = 0.5$ that is capped by a turbulent layer that stretches towards the ABL top.

Finally, the SAFIRE aircraft flew cross-sections between the wet and dry areas at 1500 m above ground level during each afternoon on 20–22 July 2021. In Fig. 4, we show the observed temperature perturbation (where the mean is taken across the entire leg) and the velocity scale defined as the square root of the instantaneous turbulent kinetic energy (uTKE $= \sqrt{u'^2 + v'^2 + w'^2}$) as used in Mangan et al. (2022a). The $x$ axis is the distance from the boundary between the wet and dry landscapes, where negative numbers are taken over the wet landscape and positive values are taken over the dry landscape. The data in Fig. 4 were taken on the flight on 22 July 2021 from 13:30 to 14:30 UTC before the arrival of the sea breeze and with relatively high values of surface fluxes. We chose to select a single flight day to highlight the structure of the turbulence and the potential turbulent coherent structures that occurred during a single afternoon. During the flight time, the wind direction was primarily from the south, with a wind speed of $5.17\,\mathrm{m\,s^{-1}}$ at the flight height near the top of the ABL, while the flight flew in a northwest–southeast transect. The data are averaged over four cross-sections of the flight that cross between the wet and dry landscapes, and data were binned by distance to the boundary for each individual transect before aggregation.

In the wet landscape, the potential temperature perturbations ($\theta'$) vary between $\pm 0.1$ K and the standard deviation in $\theta'$ is on the same order of magnitude. As the aircraft crosses into the dry landscape, the $\theta'$ increases to between $\pm 0.25$ K and the standard deviation also increases. In the dry landscape, there are temperature ramps which indicate strong buoyant activity. Likewise, the signal of uTKE shows that there is stronger turbulent transport in the dry landscape than in the wet landscape. The peaks of the uTKE signal correspond with the temperature ramps, which again suggests that the thermals are responsible for the peaks in TKE in the dry landscape. In the transition between the wet and dry landscapes, there appears to be a sudden change in boundary layer characteristics. There is an increase in $\overline{\theta'^2}$ and TKE in the wet landscape starting $\sim 500$ m before the boundary.

## 3 Refined research objectives

To aid in explaining the conclusions from the observations and to prepare hypotheses for the LES experiment, we refer to Fig. 5 throughout this section.

From the observations, we note that, above the dry landscape (right of Fig. 5), there is a prototypical convective ABL throughout the day (Figs. 2 and 3). Profiles of potential temperature and specific humidity are relatively well mixed, and, because of the warm surface, radiosonde profiles show superadiabatic lapse rates just above the surface (Fig. 2; dry profiles in Fig. 5). Likewise, the flux profiles in the dry landscape show a typical linear decrease in height in the ABL (Fig. 3). However, there is some evidence that there is some influence of the wet landscape in the dry landscape: for example, there is an increase in moisture at the top of the ABL (Fig. 2 at 08:00, 12:00, 14:00, and 16:00 UTC). Unlike the dry landscape, the wet landscape shows a layered, non-typical convective ABL (Figs. 2 and 3). During midday, the wet landscape radiosonde shows an IBL, which is suggested by the potential temperature profile (Fig. 2), and there is a local peak in buoyancy flux at approximately 25 %–50 % of the $z_i$ (Fig. 3). Above this zone, there is a turbulent layer (Fig. 3) that is well mixed (Fig. 5). Because the observations are located either in the wet landscape or in the dry landscape, we cannot say much about the interactions between boundary layers between the wet and dry areas with the observations alone.

While the observations are valuable for their high temporal resolution, the lack of spatial resolution makes it so that we cannot answer the question of how these spatial scales of surface heterogeneity interact with each other to influence the ABL development. There are three related proposed ways that the surface heterogeneity impacts the ABL (in order from left to right in Fig. 5): an advective boundary layer, an internal boundary layer, and a secondary circulation. The interaction between these impacts is a function of spatial scale and wind speed. The advective boundary layer (indicated by the arrow labeled $U$ in Fig. 5) indicates that the ABL could be formed upwind in a dry area and advected over the wet area, where it is modified, and that this modified ABL could then be advected over the dry landscape. Advective boundary layers have been shown to be a defining feature in ABL dynamics in arid regions, including the Altiplano in Chile (Aguirre-Correa et al., 2023). In this way, the wet and dry landscapes can mutually influence each other. Secondly, we potentially observe a local IBL over the alfalfa field, with the peak in buoyancy fluxes coupled with a layered potential profile; however, from the observations, we can ask if the extent of the IBL extends over the entire dry landscape scale (IBL line in Fig. 5). Finally, because the observations occur mainly in two locations, it is not clear if there is a secondary circulation that forms because of the heterogeneity (curved arrow in Fig. 5). In this way, the dry landscape can influence the wet landscape.

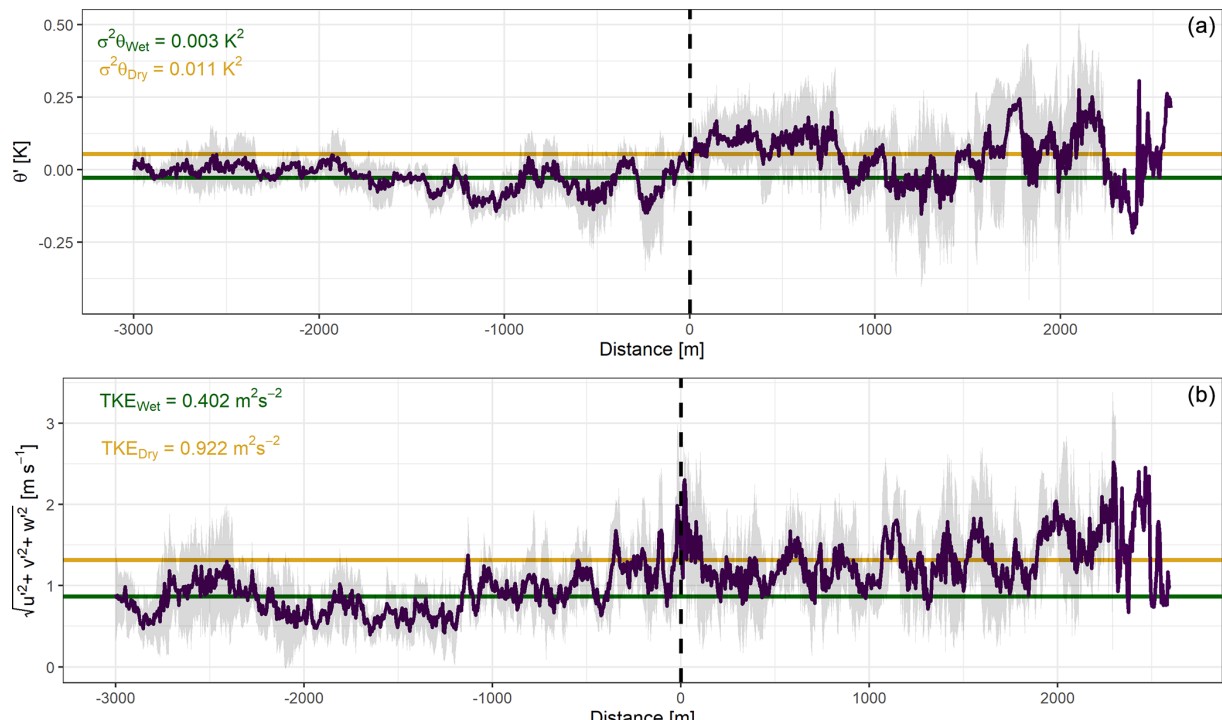

**Figure 4.** The transect from the aircraft flown on 22 July 2021 at 14:00 UTC at 1500 m a.g.l. Negative distances on the $x$ axis are in the wet landscape, and positive distances are in the dry landscape. The top panel **(a)** shows the temperature perturbations (with respect to the total transect), and the bottom panel **(b)** shows the uTKE for the same averaging time. The shading represents the standard deviation of these components averaged over four transects. The green and yellow lines indicate the average over the wet and dry landscapes, respectively (excluding $\pm 100$ m from the boundary).

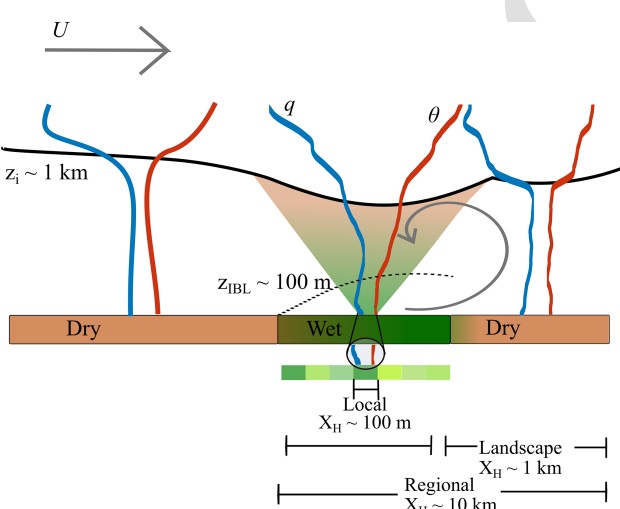

**Figure 5.** Schematic representation of the formation of the ABL (solid line, labeled as $z_i$) in the LIAISE domain. An advective boundary layer from upwind moves over the wet landscape area, where the ABL is modified with an internal boundary layer (dashed line, labeled as $z_{IBL}$). The modified boundary layer is advected over the dry landscape, where the convective turbulent motions eliminate the IBL.

To this end, we use a realistic LES to complement the observations from the LIAISE campaign. Using an LES, we can connect the wet and dry landscape scales in order to study the impact of non-local processes on the development and dynamics of the atmospheric boundary layer. We can also explicitly investigate the development of the ABL at each of the relevant spatial scales of heterogeneity. We aim to address the following research questions with the LES experiment:

1. How does the interaction between the spatial scales impact the spatial development and diurnal cycle of the ABLs in the LIAISE experiment?

2. How do the scales of heterogeneity dynamically merge in the atmosphere in space and in height?

By addressing these research questions, we aim to advance process-based understanding of how realistic unstructured heterogeneity influences a convective ABL. We can also evaluate how numerical models can capture the turbulent transport from interacting heterogeneous patches.

## 4   Large-eddy simulation

We employ the MicroHH LES (van Heerwaarden et al., 2017) using large-scale forcing downscaled to the LES do-

main (van Stratum et al., 2023) from observations of advection (Mangan et al., 2023a). In this study, we prescribe surface fluxes measured during the LIAISE campaign to ensure a realistic distribution of surface fluxes in space (Mangan et al., 2023a). In Sect. 4.1, we describe the numerical simulation, and, in Sect. 4.2, we show results for the ABL from the LES.

As our purpose for using the LES is to study the impact of the surface heterogeneity on the development of the ABL, we selected the results shown here to answer the aforementioned research questions. Therefore, we refer to Appendix B for validation of the LES compared with data from the LIAISE experiment. We also refer to the Supplement for an overview of a suite of sensitivity studies for the model configuration to the large-scale forcing and initial conditions.

## 4.1 Model configuration

MicroHH is a computational fluid dynamics simulation which supports direct numerical simulation and LES (van Heerwaarden et al., 2017). In this experiment, we employ the LES version of MicroHH at 30 m horizontal resolution over a domain of 39 km × 43 km centered on the LIAISE regional domain (Fig. 1). There are 196 grid cells in the vertical with a resolution of 25 m. Near the surface, a component the turbulent flux comes from the subgrid-scale parameterization. This is done with an eddy diffusivity closure with the Smagorinsky subgrid-scale parameterization (van Heerwaarden et al., 2017). The parameterized flux accounted for up to 5 % of the total flux from the surface to 75 m, and it fell below 1 % above 150 m. Therefore, we consider the resolved fluxes to be sufficient above this level. In the entrainment zone, the parameterized fluxes become important again, accounting for 15 % of the total flux. With this horizontal resolution of 30 m, we can capture the presence of individual fields.

We use periodic boundary conditions for the turbulent fields in the simulation. Because of this, our simulation has a "buffer zone" so that the turbulence can adjust before it reaches the inner LIAISE regional domain (Fig. 1). Based on a maximum wind speed of $\approx 4\,\mathrm{m\,s^{-1}}$ in the middle of the ABL and the $\approx 10$ km buffer around the LIAISE regional scale in all directions, the air parcel has $\approx 42$ min outside of the study area to adjust to the land surface conditions. This buffer zone should be sufficient for the turbulence to adjust to the underlying surface before reaching the area of our analysis. In this case, we use prescribed surface fluxes, the surface layer model was based on Monin–Obukhov similarity theory with no slip conditions, and there was no radiation scheme. We used a fifth-order advection scheme.

We simulated the single composite day using the MicroHH LES. We considered the first 2 h of the simulation to be spinup, so our analysis begins at 08:00 UTC. The initial profile of the atmosphere was prescribed using the radiosonde launched in the dry landscape at 06:00 UTC. In terms of large-scale forcing, we have the option to prescribe (1) advection of potential temperature, specific humidity, and wind into the domain ($U$ in Fig. 5); (2) geostrophic wind, (3) subsidence; and (4) nudging the domain mean towards observations. The advection terms for temperature, specific humidity, and wind were calculated from observations from automated weather stations located in the larger LIAISE domain. See Mangan et al. (2023a) for details of the advection calculation. Although the advection terms were calculated using 10 m wind and 2 m temperature and humidity observations, the advection terms were prescribed uniformly at all heights in the LES domain. Mangan et al. (2023a, b) show that the ERA5 reanalysis misrepresents the LIAISE domain because it does not have irrigation in this area or capture the correct location of the thermal low. For this reason, we opted to neglect geostrophic wind and subsidence. Finally, we nudged the domain-averaged profiles of potential temperature, specific humidity, and wind hourly to the radiosonde observations from the dry landscape scale.

Because our focus in this study is primarily the development of the ABL, we prescribe observed surface fluxes of sensible and latent heat fluxes and roughness lengths using the "flux map" product described by Mangan et al. (2023a). We assumed that the roughness length for momentum was 10 % of the vegetation height. By prescribing surface fluxes from observations, we can focus directly on how the atmosphere feels like a realistic surface. We are most interested in how the boundary layer forms in this study, so it is beneficial to reduce the complexity of the coupled land–atmosphere system to consider only the one-directional impact of the surface on the atmosphere.

In Appendix B, we show the validation of the LES based on the gold composite-day LIAISE profiles from the radiosondes (Fig. 2). We see that, although the model is constrained to the correct order of magnitude as the radiosonde observations are, the difference between the wet and dry local scale is smaller in the model than in the observations. This is likely because there is too much mixing in LES. We performed a number of sensitivity studies (Supplement) where we tested the influence of large-scale forcing terms on the model results. We found that the geostrophic wind is too high in ERA5, which causes a high bias in wind speed in the model compared with observations. Therefore, we opted to run the model entirely forced by observations. Furthermore, from our sensitivity study, we see that even ERA5 does not capture the larger-scale situation in the LIAISE domain well, no matter which day we chose to run. This further justifies the use of a composite-day approach. We know that the local SEBs are relatively similar across days, so all ABL variability would have to come from the larger scales. The large-scale forcings, however, exhibit large uncertainties; therefore, we cannot capture the details of each day's ABL. A composite day is then the best approach to focus on the influence of irrigation-induced heterogeneity on the ABL.

## 4.2 ABL dynamics in a realistic LES

In this section, we show the results of the LES experiment. Firstly, we analyze the spatiotemporal evolution of $z_i$ and characteristics. We consider how the ABL behaves at individual spatial scales (Sect. 4.2.1), then we shift the focus to study how the spatial scales interact with each other to influence the structure of the ABL (Sect. 4.2.2). We evaluate spatial spectra of key parameters of the ABL development to study the relevant spatial scales. We consider how the ABL adjusts as it moves from wet to dry landscape scales. Furthermore, we introduce a criterion of blending height to investigate how spatial scales blend with height in the ABL.

### 4.2.1 Spatiotemporal evolution of the ABL across spatial scales

Using Figs. 6 and 7, we can identify the spatial (Fig. 6) and temporal (Fig. 7a) variability in $z_i$ across the spatial scales of the LIAISE domain. To aid in the interpretation of ABL height in Fig. 7a, Fig. 7b has the evaporative fraction (EF $\equiv LE/(H + LE)$) as an indication of the nature of the land surface at each scale. For the LES, we defined the boundary layer height using the parcel method based on the near-surface air temperature (Stull, 1988) with a maximum jump in potential temperature of 0.25 K to identify the top of the mixed layer. Philibert et al. (2024) completed a review of ABL height methods based on observations from the LIAISE experiment. We tested the sensitivity of ABL height to the same methods and found similar (large) variability in ABL height across the methods. We proceeded with the parcel method to most closely match our analysis with the observations.

In Fig. 6, we find that the ABL is consistently higher and grows faster in the dry area than in the wet area. The height is highly variable in space. In the northwestern corner of the irrigated area, the ABL height is the lowest, and it increases towards the southeast, increasing even in the wet landscape. The maximum difference in $z_i$ between the irrigated and non-irrigated landscape is about 300 m near 12:00 UTC (Fig. 7a). However, the standard deviation within each of these scales is greater than the mean differences among them. The impact of individual fields on the local boundary layer height is greater than the total difference between the irrigated and non-irrigated landscapes. At the local scale, the ABL heights are more extreme, particularly regarding the morning growth of the ABL (Fig. 7a), but, when the sea breeze arrives by the end of the afternoon ($\text{Adv}_\theta = -0.3\,\text{K}\,\text{h}^{-1}$ and $\text{Adv}_q = -0.001\,\text{g}\,\text{kg}^{-1}\,\text{h}^{-1}$), the local scales collapse like the landscape and regional scales.

In Fig. 8, we display the spatially averaged resolved turbulent fluxes of heat (top row) and moisture (bottom row) for all spatial scales. The dashed line indicates the combined resolved and non-resolved fluxes from the between the lowest grid cells and the prescribed surface flux. The $y$ axis is normalized by the mean boundary layer height at each scale. The fallow local scale has the highest heat flux from the surface up to $z/z_i \approx 0.5$ at all times of the day. All scales have the resolved heat fluxes decreasing linearly with height to the top of the ABL, as is seen with the observations in the dry landscape from Fig. 3. The heat flux is lower on wetter scales than on drier scales. At the top of the ABL, the entrainment zone (as defined by a negative heat flux) occurs near $z/z_i \approx 0.75$ and 1.25 at all times. For heat flux, the warm air entrainment is highest at the dry scales (fallow local and dry landscape) compared to the wet scales. As in Fig. 3, the alfalfa local scale shows the increase in heat flux between the surface and $z/z_i \approx 0.2$ and weak entrainment fluxes above $z/z_i \approx 0.5$. This signature is not evident in the wet landscape scale. This suggests that an IBL forms at the local scale and not across the entire landscape scale.

All moisture in the domain comes from the irrigated fields represented by alfalfa local, which subsequently merges into the wet landscape and regional scales. In the wet local and landscape scales, there is a strong moisture flux divergence throughout the lower half of the ABL. In the dry scales, the moisture flux between the landscape and fallow local scales are low in the bottom half of the ABL. Unlike in the wet areas, there is a strong increase in moisture flux with height near the top of the ABL. This agrees with the observed moisture fluxes in Fig. 3. Near the top of the ABL, the entrainment of dry air from the free atmosphere leads to a strong moisture flux in the dry areas compared to the wet ones. Both the heat and moisture signals indicate stronger entrainment zones in the dry landscape than in the wet one, which is likely a consequence of the higher $z_i$ at these scales as shown in Fig. 7.

### 4.2.2 ABL interactions among spatial scales

So far, we have only considered the different ABLs that arise at the spatial scales of heterogeneity introduced in Sect. 2.1. In reality, these ABLs are not separate: the atmosphere mixes the impacts of the surface heterogeneity acting at different scales. To look at how the ABL mixes across the spatial scales, we start by examining the two-dimensional spatial spectra of key variables that influence the ABL development to determine their most representative length scales. This provides an indication whether the ABL reacts to the same scales of heterogeneity as the surface fluxes. Next, we study the same transect that the aircraft flew between the alfalfa and fallow fields (spatial extent of Fig. 4 and the polygon in Fig. 6c) to study how the ABL adjusts as it crosses the wet–dry boundary. Finally, we apply a model-driven approach to quantify the blending height among the spatial scales.

### Spectral analysis

Although we have defined the relevant spatial scales in this research based on physical characteristics, we can check the importance of these scales on the development of the ABL

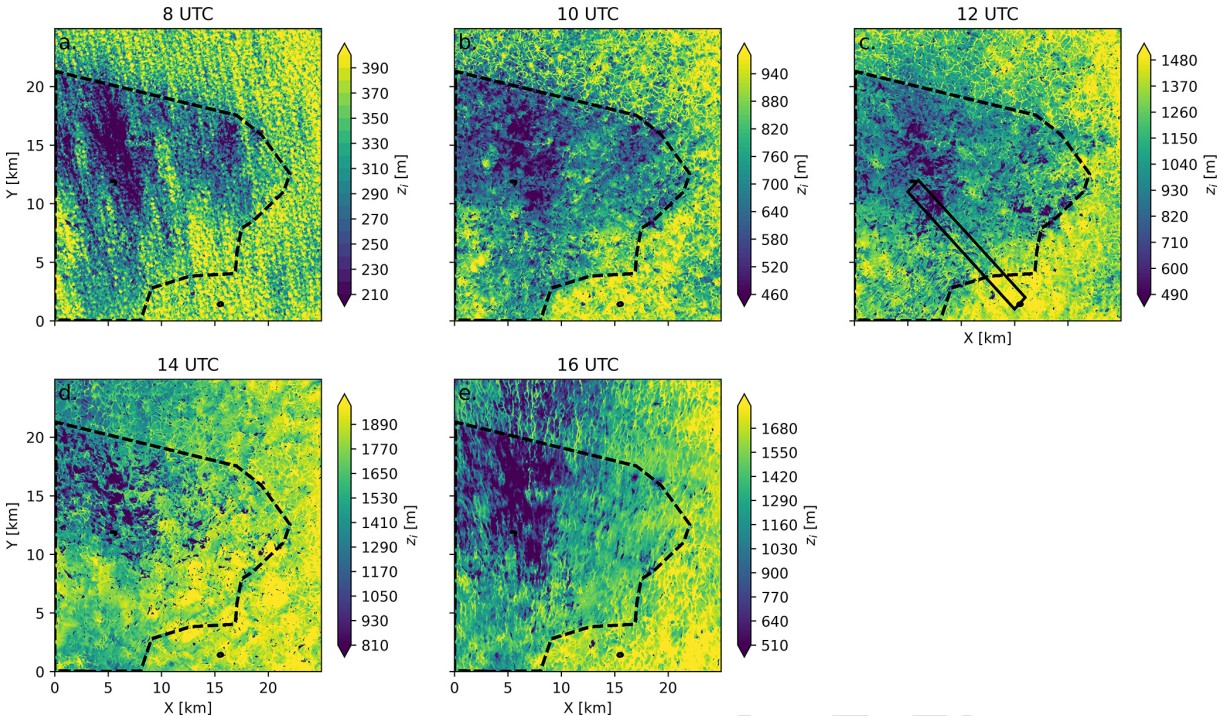

**Figure 6.** The ABL height at 08:00, 10:00, 12:00, 14:00, and 16:00 UTC calculated from the parcel method with a temperature jump of 0.25. The marked boundary is the difference between the irrigated and non-irrigated landscapes, and the points represent the alfalfa and fallow fields. The bounded box in panel **(c)** is the extent of the area used for the "adjusting ABL fluxes" from Fig. 10.

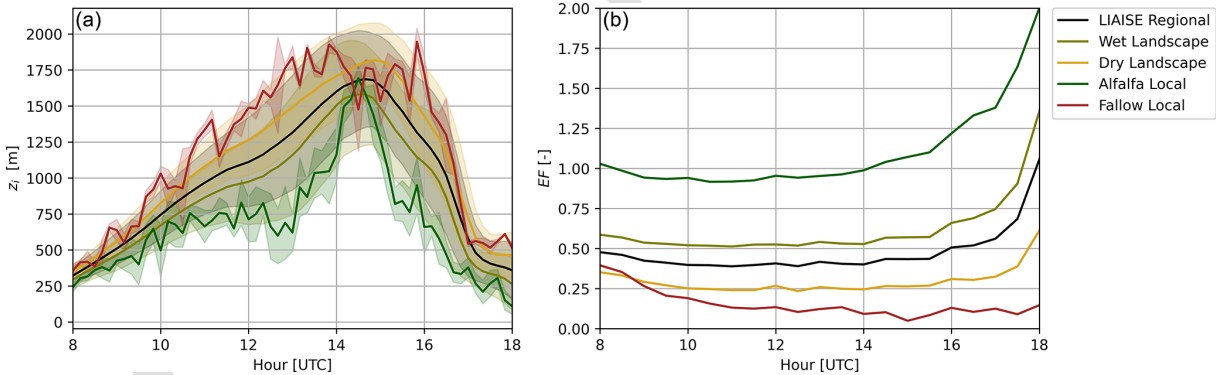

**Figure 7. (a)** The time series of the mean (solid line) and the standard deviation of boundary layer height for each spatial scale. **(b)** The mean evaporative fraction (EF $\equiv LE/[H + LE]$) at each spatial scale from 08:00–18:00 UTC.

by calculating the two-dimensional spatial spectra of surface fluxes, vertical velocity, and $z_i$ (Fig. 9). By using a spectral approach, we can identify the length scales which account for most of the variability in the signal. Moreover, we define a characteristic length scale ($\Lambda$) based on a weighted integral of the spectrum (Pino et al., 2006; de Roode et al., 2004):

$$\Lambda_\psi = \frac{\int_0^\infty S_\psi(k)k^a \mathrm{d}k}{\int_0^\infty S_\psi(k)\mathrm{d}k}; \quad a \neq 0, \tag{1}$$

where $\psi$ is a given variable, $S_\psi(k)$ is the spectral density of the variance as a function of wave number $k$, and $a$ is a weighing factor. Pino et al. (2006) describe their choice of $a = -1$ to weigh the spectra towards the large-scale ranges, while Jonker et al. (1999) chose $a = 1$ to weigh the length scale towards the smaller scales. We selected $a = -0.8$ for both the surface fluxes and $z_i$ to better capture the mesoscale peak in the spectra, while $a = -1$ was best for the vertical velocity to capture the larger scales. The characteristic length scale indicates the most important spatial scale to describe the variability in a given parameter.

Figure 9 shows the results of the spatial spectra at 12:00 UTC for (a) $z_i$, (b) sensible heat flux, (c) latent heat

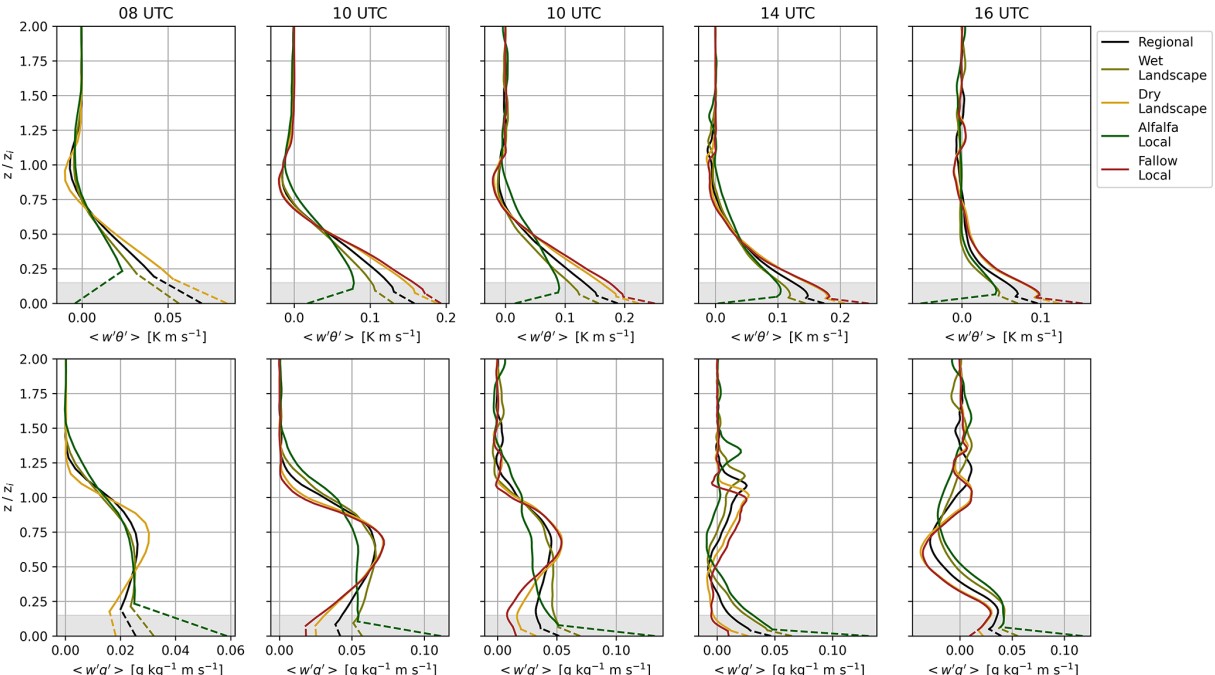

**Figure 8.** The profiles of LES-resolved fluxes with height normalized by ABL height ($z_i$) for each of the spatial scales. The shaded region indicates the surface layer. The dashed line indicates the fluxes interpolated between the lowest-resolved model level and the prescribed surface flux at each spatial scale.

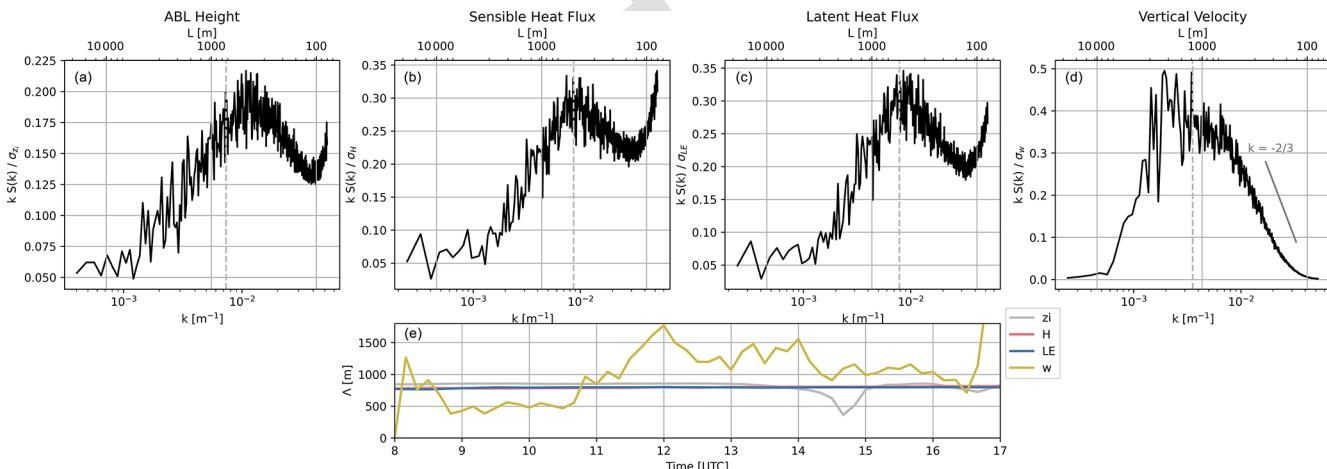

**Figure 9.** Spatial spectra computed at 12:00 UTC of **(a)** $z_i$, **(b)** surface sensible heat flux, **(c)** surface latent heat flux, and **(d)** vertical velocity at 50 % of the mean $z_i$. The dashed gray line is the location of the integral length scale for each variable at 12:00 UTC. Panel **(e)** shows the integral length scale calculated with Eq. (1) over time.

flux, and (d) vertical velocity at $0.5\langle z_i \rangle = 555$ m at the regional scale at 12:00 UTC. At this time, $z_i$ ranged from 950 m for the wet landscape scale (spectra at 0.58 % of $z_i$) to 1250 m for the dry landscape scale (spectra at 0.44 % of $z_i$). The sensible and latent heat flux spectra are from the prescribed boundary conditions. Both the surface fluxes and the $z_i$ show bimodal peaks in the spectra: one with an integral length scale of approximately 800 m and one in the microscale with a length scale of less than 100 m. The microscale peak in $z_i$

relates to the differences between individual fields. It also indicates the presence of local IBLs with characteristic length scales of $\sim 80$ m. The mesoscale peak in these signals indicates that, within the landscape scales, there is a strong variability in both surface fluxes and $z_i$. This indicates that there may be a scale between the landscape and local scales that is most important for driving the ABL growth at the regional level.

The spectrum of the vertical velocity is classical with a single peak in the mesoscale range, with a length scale on the order of 1 km, meaning that, in the middle of the ABL, the surface heterogeneity aggregates to form a circulation with a length scale of $\sim 1$ km. This relates to the same order of magnitude as the landscape scales as previously defined. Because there is little variation in the microscale spectra of vertical velocity, the influence of an IBL does not reach 50 % of the mean regional $z_i$.

Over the course of the day, the integral length scales for surface latent and sensible heat fluxes are constant (Fig. 9e). The length scale is approximately 800 m for both variables. This indicates that the sensible and latent heat fluxes are covarying. Although the evaporative fractions of different surfaces vary over the day (Fig. 7b), the ratio between LE and H stays constant in space. Like the surface fluxes, the $\Lambda_{z_i}$ is constantly $\sim 800$ m over the day, except for a brief dip between 14:00 and 15:00 UTC, where the sea breeze arrives in the domain and reduces the length scale. Finally, $\Lambda_w$ is the only variable to show a diurnal cycle, as the height at which this parameter was calculated changes over the day. In the morning, $\Lambda_w$ is $\sim 500$ m, and it increases to over 1 km at midday. When the sea breeze arrives at the end of the afternoon, $\Lambda_w$ decreases again to $\sim 1000$ m. The time-varying $\Lambda_w$ relates to the strength of the buoyancy-induced turbulent transport and has implications for the mixing between spatial scales.

### Adjusting ABL

In Fig. 10, we show the instantaneous cross-section of the LES domain between the alfalfa field and the fallow field at 12:00 UTC. The negative distances indicate the wet landscape, and the positive distances indicate the dry landscape. The transect is computed from the box in Fig. 6a that runs from the northwest to the southeast in the LIAISE regional domain (315° from north). At this time, the wind speed above the ABL is approximately 2 m s$^{-1}$, and the wind direction is predominantly from the west. Fluxes are calculated spatially over 50 transects that run parallel to the aircraft transects.

Between the wet and dry landscapes, there is an increase in $z_i$ along the transition between the wet and dry landscapes of 400 m. The ABL responds to the dry landscape downwind of the boundary (marked Distance $= 0$ in Fig. 10). There are strong updrafts within the first 500 m of the transition. If we compare this result to that of Fig. 4, we hypothesize that the aircraft may have been flying above the ABL in the wet landscape. As it enters the dry landscape, it could have flown into the ABL. The near-surface sensible heat flux increases in the dry landscape, while the moisture flux decreases. TKE is highest in the dry landscape as well. In the absence of strong horizontal winds, the TKE is concentrated in thermals, most of which occur in the dry landscape. Near the middle of the ABL, the heat flux weakens, while the moisture flux in-

creases in the wet landscape. As in Fig. 4, TKE and convection are higher in the dry landscape than in the wet landscape.

The PDFs of the fluxes confirm that, in the surface layer, the heat flux is higher in the dry area than in the wet area, and the opposite occurs with the moisture fluxes. In the surface layer, the TKE is larger in the dry landscape than in the wet landscape. For the entrainment fluxes (dotted lines), the impacts of the two landscape scales are more difficult to discern for TKE. For the heat flux, the entrainment PDFs look similar: both distributions are centered around 0. For the moisture flux, it skews negative over the dry landscape and positive over the wet landscape. For both the heat and moisture fluxes, there is a counter-gradient flux in the dry landscape. Based on profiles of potential temperature-specific humidity from the LES (Fig. B1), the along-gradient flux in the entrainment zone is warm and dry air brought from the free atmosphere into the ABL ($\overline{w'\theta'} < 0$ and $\overline{w'q'} > 0$). The counter-gradient fluxes could indicate the influence of the wet landscape on the ABL in the dry landscape. For TKE, there are pockets of strong TKE in the entrainment zone in the wet landscapes, although there is a skew to the higher TKE in the dry landscapes.

Figure 10 suggests that a weak secondary circulation forms along this transect. At approximately 2 km into the dry landscape, there appears to be a small circulation that is driven by strong updrafts. In this updraft, there is high TKE coupled with a high, positive moisture flux. This shows the presence of a thermal transporting moisture that was emitted at the surface of the wet landscape towards the top of the ABL above the dry landscape. The return flow in the wet area is not as well defined in the LES. Instead, we see lower ABL heights and some "dry tongue" pockets, as noted by van Heerwaarden et al. (2009), between $200\,\text{m} < z < 800\,\text{m}$ above the ground in the wet landscape.

### Blending height

In addition to the physical processes that are linked to the surface heterogeneity, the concept of a "blending height" arises for both numerical modeling and observational applications. In both contexts, the "blending height" describes the height at which the surface heterogeneity is no longer noticeable. We consider the blending height to be a proxy of the mixing strength of state variable and turbulent moments that arises from the surface heterogeneity. There is no consensus on the definition of blending height in literature; instead, it tends to depend on one's purpose (e.g., model or observational considerations). In Fig. 10, we see that, near the top of the ABL, the heat flux is not noticeably different between the scales; however, to quantify the areas in which the scales are blended, we use the coefficient of variation to determine the blending height, like Huang and Margulis (2009) do. The coefficient of variation ($C_{\text{v}}$) is defined as

$$C_{\text{v}} \equiv \frac{\sigma_\psi}{\langle\phi\rangle}, \tag{2}$$

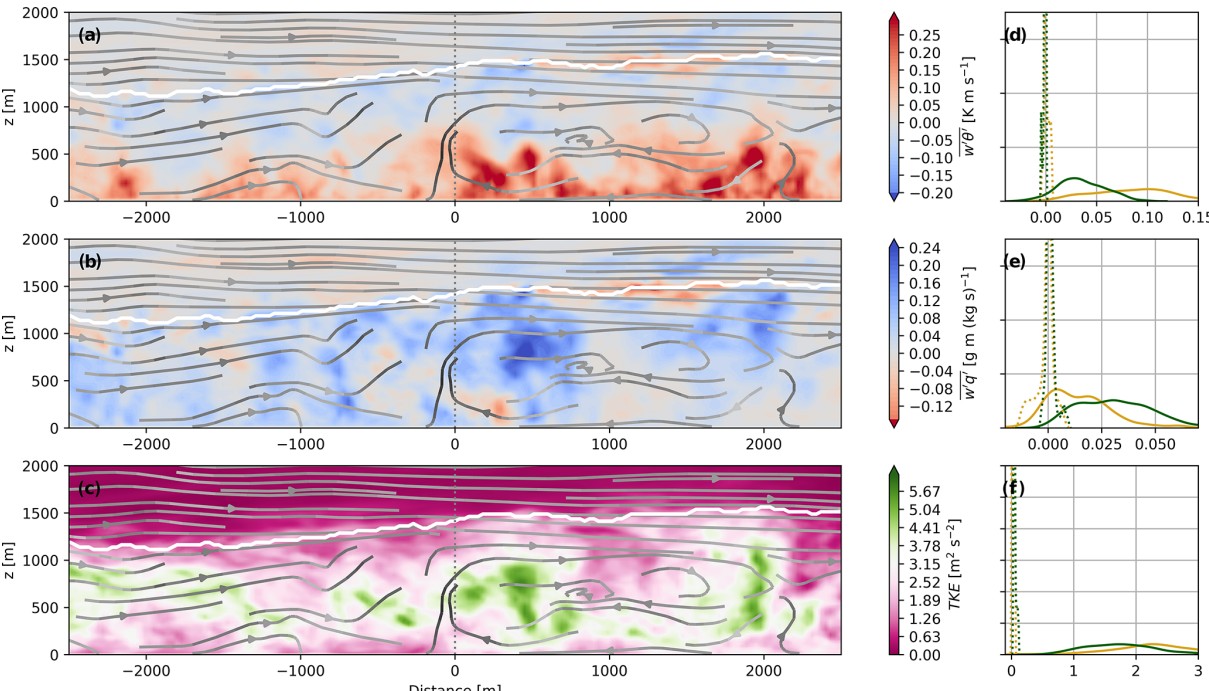

**Figure 10.** Transect from the wet landscape (negative) to the dry landscape (positive distances) from the LES. Fluxes are calculated spatially over 50 transects that run parallel to the aircraft transects. The **(a)** heat flux, **(b)** moisture flux, and **(c)** turbulence kinetic energy averaged over individual transects that mimic the aircraft strategy. The extent of the transect corresponds to Fig. 6c; negative $x$ values are in the wet landscape, and positive $x$ values are in the dry landscape. The white line is the locally determined ABL height ($z_i$). The streamlines in panels **(a)**–**(c)** are composed of the along-transect wind component and the vertical velocity. Panels **(d)**–**(f)** are the probability distributions of the fluxes from the wet landscape (green) and dry landscape (yellow) showing both the surface layer fluxes (solid lines, for $z/z_i < 0.15$) and entrainment fluxes (dotted lines, for $0.75 < z/z_i < 1.25$).

where $\psi$ is any given variable, $\sigma_\psi$ is its standard deviation in space, and $\langle\phi\rangle$ is its spatial mean. Huang and Margulis (2009) defined the blending height as the level in which the coefficient of variation for the heterogeneous case is less than or equal to that of the homogeneous case. However, a blending height can be defined relative to a number of dimensions, including location, height, and domain. Therefore, in a general form, the unitless blending parameter ($B$) can be expressed as the ratio of $C_v$ of dimension to that of another,

$$B(x, y, S, z) \equiv \left| \frac{C_{v,1}}{C_{v,2}} \right| \tag{3}$$

where $x$ and $y$ are locations in space, $z$ is height, and $S$ is spatial scale (e.g., domain or model resolution). In its most general form, the blending height expresses the lowest level in the atmosphere where the surface heterogeneity is not felt by the atmosphere. In that case, the appropriate form of Eq. (3) would take the form

$$B(z) = \left| \frac{C_v(z)}{C_v(z = 0)} \right|. \tag{4}$$

$C_v(z = 0)$ would take the variable averaged at the surface over the entire heterogeneous domain, while $C_v(z)$ would be

taken as a function of height and while the scale is held constant. With this definition, we observe a local minimum of $B$ inside the ABL, which represents a blending zone, and a local maximum of $B$ in the entrainment zone corresponding to the entrainment processes, which causes the atmospheric fields to be heterogeneous. This implies that the surface heterogeneity blends towards the top of the atmosphere ($z/z_i \approx 0.7$ for potential temperature) but that entrainment processes reintroduce atmospheric heterogeneity at the top of the ABL.

The general form can also potentially be extracted in space to be applied as a blending distance used in plume dispersion measurements as introduced by Schulte et al. (2022). To do so, one would hold $S$ and $z$ constant and compute the blending distance by altering the locations in space $x$ and $y$. In that case, it could take the form

$$B(x, y) = \left| \frac{C_v(x, y)}{C_v(x, y = 0)} \right|, \tag{5}$$

where $C_v(x, y = 0)$ could be the background variability.

In this case, however, we are interested in the height where the spatial scales ($S$) blend together. By comparing scales directly, we can study the impacts of the heterogeneity on the grid size of a regional or global model. Therefore, we define

the blended zone to be where the $C_v$ values of two scales are within 5 % of each other. We found that the choice of threshold was not sensitive to the results of the blended zone for thresholds between 2 % and 10 %. In effect, this method means that we consider scales to be blended if the normalized variability in a given variable is approximately equal. In specific form, the blending definition used in this study is TS1

$$B_{S1 \to S2}(z) \equiv \left| \frac{C_{v,S1}(z)}{C_{v,S2}(z)} \right| \leq 0.05. \tag{6}$$

In Fig. 11, we show a time series of the blending height from the surface (normalized by $z_i$ for the smaller scale) for the scalars of potential temperature and specific humidity and the fluxes of heat and moisture for each of the scales. Because our scales are nested, we analyze the height at which the local scales (alfalfa and fallow) blend into their respective landscape scales (wet and dry) in Fig. 11a and b. We can also study where the landscape scales blend into the total LIAISE regional scale in Fig. 11c and d. We do not identify a strong diurnal cycle for these blending heights, and the lowest level of blending depends on the selected variable.

The local scales blend into the landscape scales within or closely above the surface layer. At the fallow scale, the moisture flux blends between $0.3 < z/z_i < 0.4$ in the morning, while all other fluxes blend within the surface layer ($z/z_i < 0.15$). The alfalfa local scale blends at a higher height for potential temperature and heat flux than the moisture variables. In the morning, the blending height for moisture terms is higher than that of the heat terms. When we compare this result to the flux profiles in Fig. 8, we observe that, between alfalfa local and wet landscape scales, the moisture fluxes converge closer to the surface than the heat fluxes and that, between the dry landscape and fallow local scales, the heat fluxes converge closer to the surface than the moisture fluxes. This might suggest that the scales mix closer to the surface for variables which are more similar between scales; the dominant process for a given scale has a lower blending height.

Unlike the local scales, which blend low in the ABL, the landscape scales remain different from the regional scale until the top of the ABL. Scalars such as potential temperature and specific humidity blend from $0.6 < z/z_i < 1.0$ in the morning, while fluxes blend lower: heat flux blends in the surface layer, and moisture flux increases over the day between $0.2 < z/z_i < 0.8$.

In almost all cases, except for the alfalfa local scale, which is characterized by near-zero or negative heat flux, the heat flux blends in the surface layer. This is likely because the heat flux is the dominant component of the buoyancy flux, which is controlling the mixing in this convective boundary layer case. However, the moisture flux does not blend at the same locations. This indicates that the mixing process is not physically the same between heat and moisture. It supports previous research that suggests there is dissimilarity in turbulent transport between heat and moisture fluxes (e.g., Huang

et al., 2009). The scalar values do not mix until the top of the ABL in the morning, but, in the afternoon, when the domain becomes more convective, the ABL becomes better mixed in the scalars.

Another potential reason for the difference in blending heights between the heat and moisture variables is that the distributions of the specific humidity and moisture fluxes are more heterogeneous than those of the temperature. To illustrate this, we consider a scale analysis. The mean mixed-layer potential temperature $\sim 300\,\text{K}$ and perturbations from the mean are $\sim 1$–$2\,\text{K}$ depending on the location in the mixed layer, so the $C_v \approx 0.5\,\%$. Conversely, specific humidity has a mixed-layer mean of $\sim 10\,\text{g kg}^{-1}$ with perturbations of $\sim 1\,\text{g kg}^{-1}$, so the $C_v \approx 10\,\%$. By normalizing the $C_v$ in Eq. (3) by a reference $C_v$, the relative magnitude of the $C_v$ is taken into account. However, even with this normalization, it still holds that small changes in humidity cause a more heterogeneous atmosphere in terms of humidity than small changes in temperature.

It is important to note not only that the blending differs based on the variable of interest but that the entrainment zone causes variables not to be blended. This is because entrainment rates are not uniform across the spatial scales. The difference in entrainment introduces heterogeneity into the ABL from the top. It implies that, although, in the middle of the ABL, statistically, the surface is not directly felt, the process of entrainment still feels the surface and the ABL itself is heterogeneous. This could be related to the presence of thermals preferentially in one region (e.g., dry landscape). The transport may not be statistically different within and outside of thermals, but strong updrafts could push up the ABL top leading to more entrainment. Thereby, like the surface, the entrainment zone "de-blends" the flow in heterogeneous areas. We show an example of this in Appendix C.

## 5 Discussion

Surface heterogeneities can impact the ABL in a number of ways both vertically and horizontally in the ABL. In this study, we use the realistic case from the LIAISE campaign and a combination of observational experiments and a numerical experiment using LES. In order to combine the methods, we firstly discuss the dynamics of the ABL across the spatial scales in Sect. 5.1. In Sect. 5.2, we shift our focus to the implications for how subgrid-scale heterogeneity is handled in numerical models.

### 5.1 ABL across scales

At the local scales (the alfalfa and fallow fields), we find that the land surface is not large enough to impact the entire depth of the ABL. There is high variation in $z_i$ and characteristics such as temperature and humidity, which indicates the presence of localized IBLs at this scale. From observations, there is a clear IBL at the alfalfa local scale based on the layered

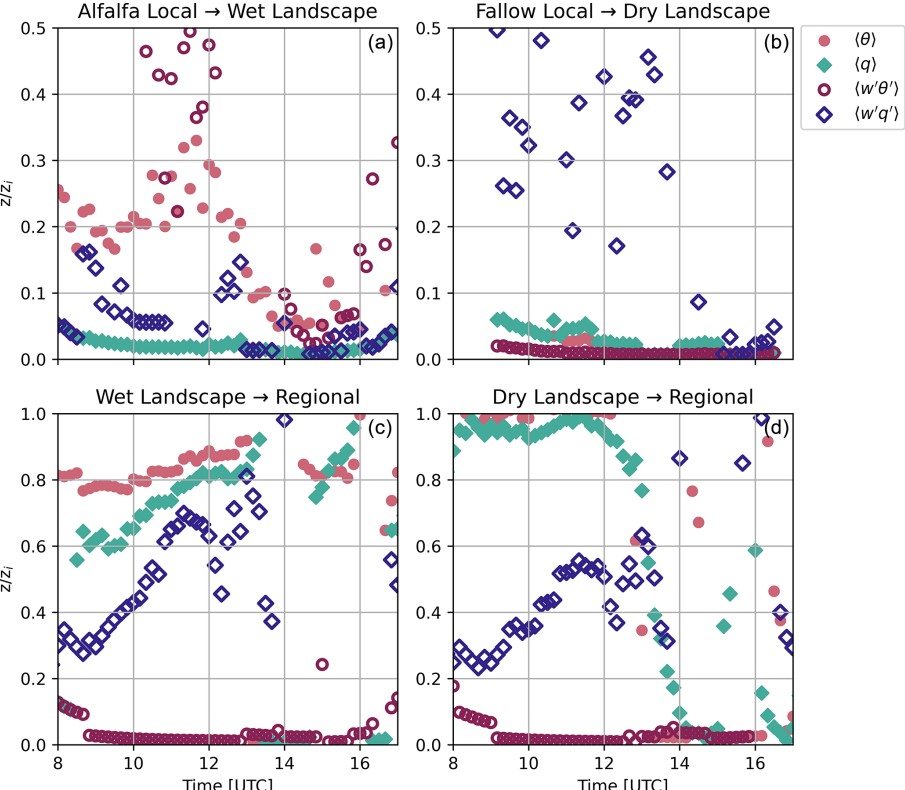

**Figure 11.** The surface blending height with respect to time for **(a)** the alfalfa local scale blending into the wet landscape scale, **(b)** the fallow local scale blending into the dry landscape scale, **(c)** the wet landscape scale blending into the LIAISE regional scale, and **(d)** the dry landscape scale blending into the LIAISE regional scale. The colors are the blending of the variable: potential temperature (pink), heat flux (brown), specific humidity (green), and moisture flux (blue).

potential temperature profile of the radiosonde and the flux regime in the bottom half of the ABL. At the alfalfa local scale, the surface layer becomes stable, but it is topped by a convective boundary layer (Fig. 2). The LES is able to cap-
5 ture the stable layer at this scale as shown through the flux profiles (Fig. 8). At the fallow local scale, there is not a clear IBL from either the observations or the LES because it is relatively warmer than the domain. Finally, the peak variability in this length scale in the spectra of $z_i$ indicates that IBLs are
10 formed at this scale but that IBLs do not form at the landscape scale.

The wet and dry landscape scales are characterized by the differences between fields of the irrigated and non-irrigated areas of the LIAISE domain. From the LES, we find that the
15 mean $z_i$ of the wet landscape is at most 300 m lower than that of the dry landscape. The spectral analysis of $z_i$ shows that it varies most within the landscape scale ($\Lambda_{z_i} \sim 800$ m). At the landscape scale, the variability in the height of the ABL is larger than the mean differences in the height of the ABL.
This indicates that the local scales which make up the landscape scale are more variable within each landscape scale than between them. The local scales blend into the landscape scales within the surface layer of the ABL, so the impacts

of individual fields do not impact the ABL above $z/z_i \approx 0.3$. In Fig. 10, we find that there is a gradual increase in $z_i$ in
a cross-section between the landscape scales and that TKE enhances the mixing in the dry landscape scale compared to the wet landscape scales. Finally, there is no evidence of secondary circulations within each landscape scale because the length scale of heterogeneity is smaller than the $z_i$, but there
is indication of a circulation forming between the wet and dry landscape scales. This agrees with findings from Patton et al. (2005).

The regional scale is characterized by the heterogeneity between the wet and dry landscape scales. We might expect
a secondary circulation to occur within this scale based on the scale of the heterogeneity and its intensity (Patton et al., 2005). In the LIAISE area, Lunel et al. (2024a) simulated a well-defined secondary circulation with a mesoscale model; however, our results do not show as strong a secondary cir-
culation. There are a few possible reasons for this. The first potential reason is related to the pattern and strength of the heterogeneity. In our case, because the heterogeneity is relatively unstructured in the landscape scale, its impacts are not felt as strongly as if they were structured. In this study,
we focused on a smaller spatial extent, while Lunel et al.

(2024a) modeled all of Catalonia and found secondary circulations with length scales of $\sim 100$ km. Furthermore, they used a coupled model where the soil moisture in the irrigated fields was at field capacity, which exaggerated the differences between the wet and dry landscapes compared with our study. In this study, we maintained the microscale heterogeneity that occurred within the landscape scale, which, on the whole, lessened the differences in surface fluxes between the wet and dry landscape scales. The second reason for a weakened secondary circulation is the presence of a background wind. Hechtel et al. (1990) ran an LES with a realistic land surface and found no secondary circulation, which they suspected was due to the high velocity in the domain. Avissar and Schmidt (1998) and Raasch and Harbusch (2001) found that a mean background wind greater than $2.5\,\mathrm{m\,s}^{-1}$ reduces the formation of secondary circulations if the wind is not normal to the boundary. In our case, the wind is mainly westerly, but the boundary is curved.

Unlike the landscape scales, the blending height from the surface in the regional scale occurs between $0.3 < z/z_i < 0.8$ in the mornings before the atmosphere is convective. During the afternoon convective period, the blending heights decrease below $0.2 z/z_i$ for potential temperature and for the heat flux. However, the heat and moisture fluxes do not blend until near the top of the boundary layer even when the atmosphere has strong, convective mixing. Convective turbulence, which arises from the buoyancy flux, is responsible for controlling the blending height. Moreover, because the entrainment varies between the wet and dry landscapes, we observe that there is an "unblended" zone at the top of the ABL in the regional scale. This implies that, although heterogeneity arises from the surface, its impacts are felt through the entire ABL.

## 5.2 Implications for handling subgrid heterogeneity

In our study, we focus on the variety of physical processes that control the dynamics of the ABL. Despite our focus on process understanding, the results of this study could have implications on how subgrid-scale heterogeneity is handled in regional-scale weather models. Subgrid-scale surface heterogeneity is often handled either with the parameter aggregation approach or with the flux aggregation approach. In the parameter aggregation method, the land surface is linearly averaged over the heterogeneity. Mangan et al. (2023a, b) use a parameter aggregation method to represent the LIAISE experiment. Conversely, in this study, we prescribe the surface based on observations, which allows the atmosphere to mix out the impacts of the heterogeneity and allows this approach to be a proxy flux aggregation approach.

Mangan et al. (2023a) found that the parameter aggregation over the regional and landscape scales re-created the dynamics of the ABL from the observations. This suggested that the regional ABL was formed through a combination of the land surface from both the wet and dry areas. In this study, where scales are able to interact in the atmosphere, we notice similar results: the difference in the $z_i$ between the regional scale comprises a composite of the landscape scales, and its impacts are relatively linear. Individual fields have little impact on the regional scale; however, collectively, the relatively wet and dry landscape scales begin to impact the regional scale. More research needs to be done to quantify the influence of the altered ABL on surface fluxes and how that should best be quantified in numerical models.

The regional scale represents a single grid cell of a current global model, and the selection of the landscape and local scales represents possible future model resolutions where the sources and strength of subgrid-scale heterogeneity differs. At the resolution of regional scale, flux aggregation occurs at $z/z_i \approx 0.5$; however, it varies depending on both the variable and the atmospheric stability. The differences in fluxes in the entrainment zone should be taken into account in order to better capture the impacts of the surface heterogeneity on the ABL. We find that the differences between flux profiles and ABL height are more notable within the landscape scales than between them. From the surface, the blending between the local and landscape scales occurs in the surface layer, so the aggregation occurring near the surface is more reasonable at this resolution.

While, in this case, we find that the blending height from the surface depends on both variable and spatial scale, further research should be done to investigate how generalizable this result is in both geographical location and season. We hypothesize that this multi-scaled approach is reasonable in other irrigated semi-arid regions because of typical irrigation patterns. However, as we mentioned in the formation of the LES experiments, the mesoscale forcing is vital for correctly capturing the correct mixing between the heterogeneous land surfaces. In regions with difference synoptic and mesoscale circulations, the influence of how the surface heterogeneity influences the blending in the ABL could vary.

## 6 Conclusions

By combining surface and upper-air observations of mean and flux-state variables of the ABL with a high-resolution, realistic LES, we study the dynamics of the ABL. Particular emphasis is placed on how these dynamics act across different spatial scales driven by a very large surface heterogeneity with characteristic length scales. The observations show the presence of an IBL locally in an alfalfa field, where buoyancy flux at the surface is solely driven by the moisture flux. In contrast, they show a prototypical convective ABL in the fallow field. We hypothesize that there are interactions between the irrigated and non-irrigated areas which influence the dynamics of the ABL. Our main findings are the following:

1. *How does the interaction between the spatial scales impact the spatial development and diurnal cycle of the ABLs in the LIAISE experiment?*

The local scales, which are characterized by individual fields, are driven by extreme surface fluxes ($\beta < 0.1$ and $\beta > 20$). IBLs form at this scale over the course of the day, but they do not persist at the larger scales. Because of the relative size of the local scale ($\sim 100$ m) compared to the $z_i$ ($\sim 1000$ m), the impact of the local scale on the total ABL is relatively limited. Therefore, in both the model and observations, we find a stable surface layer topped by a convective boundary layer in the alfalfa local scale. At the landscape scales, the heterogeneity arises from individual fields within the irrigated or non-irrigated areas. The variability in the ABL within the landscape scales is much larger than the differences among the scales; therefore, our findings show that there is no IBL that forms within or between the landscape scales. From the LES numerical experiments, both landscape scales (wet and dry) show a prototypical ABL, although buoyancy is weaker in the wet landscape than in the dry landscape. Finally, at the regional scale, heterogeneity is formed by the contrast between irrigated and non-irrigated agricultural fields. The regional ABL characteristics fall between the extremes of the two landscape scales, which could be a function of the strong vertical mixing that arises from the dry landscape scale.

2. *How do the scales of heterogeneity dynamically merge in the atmosphere in space and in height?*

Based on spectral analysis of the LES results, we observe that the length scale of the ABL height follows that of the surface fluxes. There is peak variability in ABL height in the mesoscale ($\Lambda_{z_i} \sim 800$ m), which relates to the landscape scale, and in the microscale ($\sim 100$ m), which confirms that IBLs are formed at the local scale. By analyzing the blending height from the LES, we discover that the local scales blend into the landscape scales by $z/z_i = 0.3$ for scalars and turbulent fluxes under convective conditions. The landscape scales blend into the regional scale by $z/z_i = 0.8$ for scalar turbulent fluxes under convective conditions. The blending height also depends on the variable of interest and the strength of the buoyancy of the turbulence. Generally, moisture variables blend higher in the atmosphere than temperature variables because moisture is more heterogeneous in the ABL than temperature. Because of the surface heterogeneity, there is also spatial heterogeneity in entrainment fluxes, which leads to a second, unblended layer in the entrainment zone above $0.8 \, z/z_i$. The role of entrainment fluxes in blending within the ABL should be taken into account in weather models to better capture the impacts of surface heterogeneity on the ABL.

Increasing our understanding of how surface heterogeneity impacts the dynamics of the ABL in a realistic case is an important step towards understanding the full impacts of surface heterogeneity on the bi-directional land–atmosphere interactions. As opposed to previous LES studies which created ABL scaling for heterogeneous surfaces, we apply a realistic land surface and atmospheric conditions to understand how these concepts hold with realistic configuration of surface heterogeneity. In the LIAISE domain, the local field scales mix into the landscape scales close to the surface, while the impacts of the landscape scales are felt throughout the depth of the ABL and with a horizontal length scale $\sim 800$ m. Because of the patchy surface representation and a maximum background wind of $2 \, \mathrm{m \, s^{-1}}$, the impacts of secondary circulations are not felt as clearly as previous idealized LES studies suggest (e.g., Patton et al., 2005; van Heerwaarden and Vilà-Guerau de Arellano, 2008). Finally, by combining observations with an LES case study, we can test how well we can capture the observed ABL in a heterogeneous region. We found that properly constraining the meso- and synoptic-scale forcing is vital for determining the influence of surface heterogeneity on the ABL.

## Appendix A: Observed boundary layer height

**Table A1.** The average observed boundary layer height ($z_i$) and internal boundary layer height ($z_{IBL}$) calculated from the radiosondes from the wet and dry landscape scales.

| Time (UTC) | Wet landscape | | Dry landscape |
| | $z_i$ (m) | $z_{IBL}$ (m) | $z_i$ (m) |
| --- | --- | --- | --- |
| 08:00 | 268.51 | 202.02 | 595.00 |
| 10:00 | 425.59 | 217.39 | 843.67 |
| 12:00 | 631.70 | 148.37 | 1143.00 |
| 14:00 | 1236.36 | 143.90 | 1497.33 |
| 16:00 | 1029.53 | 130.65 | 1119.67 |

## Appendix B: Validation of the LES experiment

We performed sensitivity studies for this LES experiment with the large-scale forcing components. In Fig. B1, we show the results of the spatial means of potential temperature and specific humidity for the LES scales compared to the radiosondes with LIAISE observations. We expect the radiosondes to best represent the wet and dry landscape scales. The shading represents the standard deviation of the observations averaged over the LIAISE composite days.

In our selected case, the LES captures the approximate ABL height compared to the radiosondes for both the wet and dry landscapes. In the afternoon, there is a warm bias compared to the radiosondes at all scales, but moisture appears to be well captured. Furthermore, the spread between the wet and dry landscape scales in the model is less than with observations. This is likely due to the surface representation and the assumptions used, as we were able to capture the observed wind using the observationally driven large-scale forcing terms.

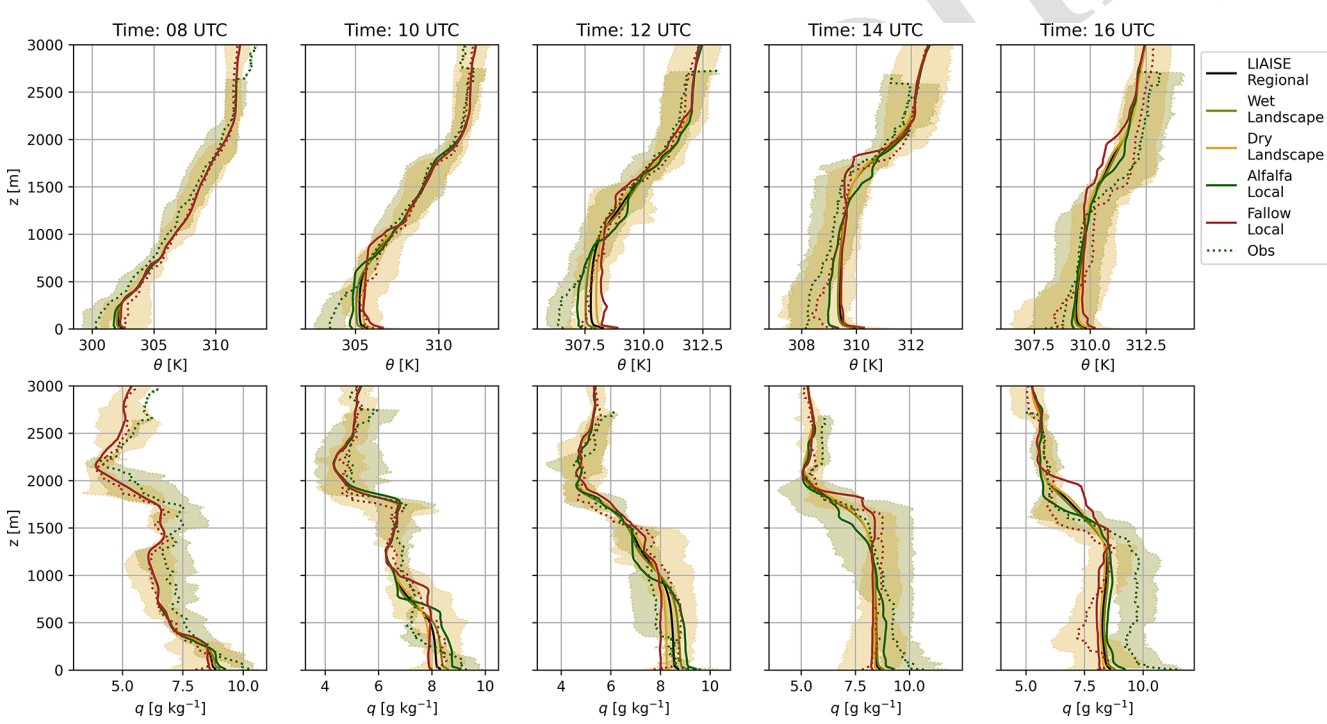

**Figure B1.** The LIAISE composite-day radiosondes (dotted line and shading) with the spatial averages of the potential temperature and specific humidity from the LES. The shading is the daily range in the observations as shown in Fig. 2. The solid colored line corresponds to the different spatial scales.

## Appendix C: Blending height: additional results

Because blending height was computed spatially, we can observe the "blending region" in the ABL using a time–height figure. Figure C1 shows an example of the "blended zone" from the wet landscape to the LIAISE regional scales for (a) potential temperature, (b) specific humidity, (c) heat flux, and (d) moisture flux. The gray indicates the blended zones (where $\left|\frac{C_{v,\,wet}}{C_{v,\,dry}}\right| < 0.05$), and white indicates the non-blended zones. The black and green lines indicate $z_i$ of the regional and wet landscape scales, respectively. The method of determining the blending height has errors when the $C_v$ approaches zero, meaning the variable is homogeneous in space. For that reason, the method does not work well above the ABL.

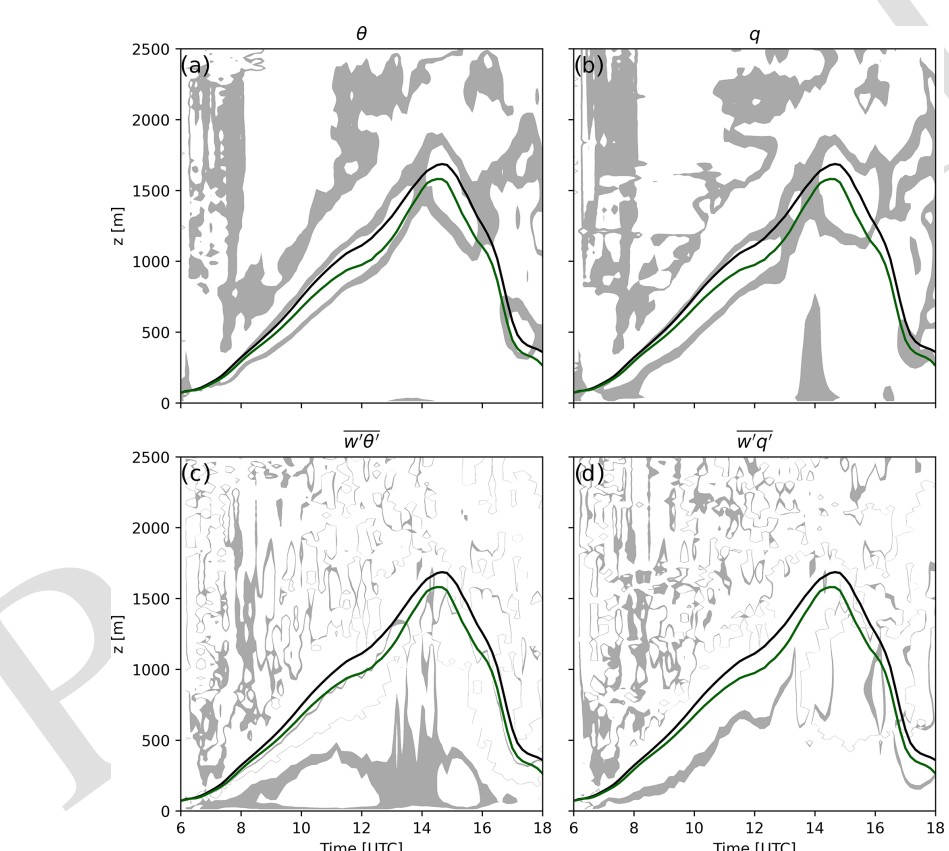

**Figure C1.** Time–height figures for blended variables between the wet landscape and the LIAISE regional scales. The gray areas indicate that the scales are blended at a certain time. The variables are **(a)** potential temperature, **(b)** specific humidity, **(c)** heat flux, and **(d)** moisture flux.

**Code and data availability.** In situ observations from the LI-AISE field experiment can be found in the LIAISE catalog: https://liaise.aeris-data.fr/ (last access: TS2; DOIs: https://doi.org/10.25326/528, Canut, 2023; https://doi.org/10.25326/322, Garrouste and Canut, 2022; https://doi.org/10.25326/365, Lothon and Canut, 2022; https://doi.org/10.25326/391, Mangan et al., 2022b; https://doi.org/10.25326/390, Mangan et al., 2022c; https://doi.org/10.25326/429, Price, 2023). The MicroHH LES code can be found at https://github.com/microhh/microhh (van Heerwaarden et al., 2017), and its documentation can be found at https://microhh.readthedocs.io/en/latest/index.html (last access: 2 July 2025). The statistics from the LES experiment and the input data are available for download at https://doi.org/10.5281/zenodo.13379335 (Mangan et al., 2024).

**Supplement.** The supplement related to this article is available online at [the link will be implemented upon publication].

**Author contributions.** MRM: conceptualization, methodology, formal analysis, writing (original draft and preparation). JVGdA: conceptualization, supervision, writing (review and editing). BJHvS: methodology, software, writing (review and editing). ML: data curation, methodology. GCR: data curation, methodology. OKH: conceptualization, supervision, writing (review and editing).

**Competing interests.** The contact author has declared that none of the authors has any competing interests.

**Disclaimer.** Publisher's note: Copernicus Publications remains neutral with regard to jurisdictional claims made in the text, published maps, institutional affiliations, or any other geographical representation in this paper. While Copernicus Publications makes every effort to include appropriate place names, the final responsibility lies with the authors.

**Acknowledgements.** We would like to thank the hosts, organizers, and participants of the LIAISE campaign. Kyaw Tha Paw U and Donold Lenschow provided interesting discussions which influenced sections of this study. We would also like to thank the three anonymous reviewers and Dennis Baldocchi for their detailed feedback which greatly improved this study.

**Financial support.** This PhD project was partly supported by the appointment of Jordi Vilà-Guerau de Arellano as Chair of the Meteorology and Air Quality Group of Wageningen University TS3. TS4

**Review statement.** This paper was edited by Michael Byrne and reviewed by three anonymous referees.

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

## Remarks from the typesetter