# Peer review of "The spatiotemporal evolution of atmospheric boundary layers over a thermally heterogeneous landscape"

_EGUsphere, 2024_

## Referee Comment (RC2)

This manuscript combines surface and radiosonde observations of the atmospheric boundary layer (ABL) with large eddy simulation (LES) experiments to investigate the role of surface heterogeneity on the dynamics and thermodynamics of the ABL at a variety of spatial scales. The topic is interesting and timely for the ABL community, and the manuscript presents some interesting results, particularly on the interplay of advection and entrainment processes in shaping the ABL. However, I find the manuscript to be poorly structured and poorly written to the extent it makes it difficult for the reader to follow (some suggestions below). I highly recommend a major overhaul to the structure/writing. I also have some technical comments that I am hoping would help improve the manuscript.

1) Figure 2: The authors correctly discuss that the upper part of the ABL above the wet surface (Alfalfa) becomes drier due to advection or entrainment in the early afternoon. However, given that the temperature profile in this part still shows a colder ABL, does this suggest that advection is the main reason? Ideally, free tropospheric air would be drier and warmer than the ABL air, so entrainment would dry and warm the ABL. Given the wide range of uncertainty in the figure (shaded regions), it may be worthwhile conditioning this analysis on horizontal wind speeds to characterize advection.

2) Figure 3: the buoyancy flux near the surface seems smaller than the kinematic heat flux, on average. Is this just a visual issue. This needs explanation.

3) Line 178-179: This is a technical comment on turbulence isotropy. I suppose the authors mean component wise isotropy not turbulence isotropy, because the variances $\overline{u'^2}$, $\overline{v'^2}$, and $\overline{w'^2}$ can be equal and turbulence remains anisotropic in the Kolmogorov sense. Also, the units of the variances are m$^2$ s$^{-2}$ not s$^{-1}$

4) Line 297 and Figure 6: It is difficult to make sense of the fact that the variability in $z_i$ within a patch (irrigated vs. non irrigated) is larger than the mean difference between the patches. Can the authors show the map of surface fluxes prescribed in LES? Are these fluxes homogeneous over a patch? This may hold some clues.

5) Line 202: The text refers to potential temperature $\theta'$ and the figure (4) shows absolute temperature $T$. Which is it?

6) Figure 4: Given tha Primes are defined as fluctuations around the full transect, shouldn't the sum of $T'$ averaged over the wet and dry areas be zero? Similarly for uTKE. The horizontal green and yellow lines do not seem like it.

7) Line 385: what is meant by "counter-gradient flux" in this context? This is not clear unless the temperature and humidity profiles are themselves shown.

8) Figure 10: stronger TKE pockets over the dry region (right part of domain) are accompanied with higher latent heat fluxes over that region, which is interesting. I don't see this discussed well, particularly in the context of a secondary circulation.

9) Figure 11: A blending height of $0.8z_i$ (wet landscape) means that there is effectively no blending (i.e., heterogeneity effects reach far up in the ABL). Also, the difference in blending height between temperature and humidity in Fig. 11a (Alfalfa) is puzzling. Even if these have some sort of transport dissimilarity as the authors argue, such a big difference suggests that some physical mechanism is at play that was not discussed, or perhaps a consequence of the method the authors use to estimate blending height?

10) Abstract Lines 9 and 10: "near 1000 m" and "~500m" are vague. Please clarify whether these represent horizontal scales or heights in the ABL.

11) Abstract Line 11: I suggest replacing "three-dimensionality of LES" with "spatio-temporal extent of LES"
12) Line 47: "aggregating the land surface properties"

13) Line 93: "Fig. 1 shows ..."
14) Line 126: replace "... temperature and humidity in a profile" with "temperature and humidity profiles".

15) Line 145: "we observe ...". This sentence is unclear; consider revising.
16) Line 283: remove "at"
17) Line 290: Fig. 7 not 7 7
18) Line 301: K/hr
19) Line 303: add a period after scales.

---

## Author Response (AR1)

**Response to Editor**

Mary Rose Mangan

Wageningen University

Meteorology and Air Quality Group

Droevendaalsesteeg 3a

 6708 PB Wageningen, the Netherlands

Dear Dr. Byrne,

Thank you for your consideration in reviewing our paper titled "The spatiotemporal evolution of atmospheric boundary layers over a thermally heterogeneous landscape" for the journal Atmospheric Chemistry and Physics. We also thank the three reviewers and Dr. Dennis Baldocchi, as all of the reviews were thorough and raise important issues that help to improve the science, scope and interoperability of the paper. We have incorporated the suggestions of the reviewers into a revised manuscript.

Generally, the reviewers raised two major points: the representativity of the composite golden day (and the drawbacks of using such an approach) and better explaining the uncertainties and physical interpretation of the blending height. Throughout the comments, we have addressed these concerns Because we had selected an unusual structure of the paper, two of the reviewers gave recommendations on how to clarify this for the readers. Based on all of these comments, we have made major revisions of the manuscript.

We have edited the manuscript to address the comments by the reviewers. Attached to this letter is the response to the reviewers and the community comment. All the comments by the reviewers can be found below in red font. Our point-by-point response is indicated black, non- italicized font. Any proposed textual changes that we have included in the response to the reviewers, we have written in black italicized fonts.

Sincerely,

Mary Rose Mangan

Reviewer #1

**General comments**

This article presents a study of the atmospheric boundary layer over a heterogeneous surface. To do this, the first part of the article uses observational data from a field campaign, and the second part runs and explores a LES on the area of the same campaign. Overall, the article is well written. The article gives interesting results on the ABL dynamics over a heterogeneous surface and in particular on the blending height of atmospheric state variables. However, some shortcomings in the methodology affect the quality of the article. In particular, the analysis of a composite day created by averaging 3 radiosoundings seems questionable. In my opinion, it prevents the authors from fully understanding the meteorological situation they are trying to study and makes the comparison with observations more difficult. Some results (such as the presence of an IBL) are not based on objective quantification but on subjective judgement. The article is relatively long, with weak links between section 2 (observations) and section 4 (LES), and the structure is sometimes confusing. In addition, some results could be discussed in a more balanced way, with more discussion of the limitations of this work. I think Appendix A is important as it shows that the LES does not capture the observed ABL dynamics, which is probably the reason why Sections 2 and 4 are not so well connected.

Thank you for your very thorough comments and your thoughtful suggestions. Your comments have been extremely useful in revising the manuscript. We understand your comments to be centered around two major points: (1) the connection between sections 2 and 4 including the validation of the LES with observations, and (2) the use a composite day. In the specific comments, you commented on both of these issues in multiple locations, so we will summarize our response here, but we have saved the detailed response on specific points to those comments.

As you suggested, more validation of the model compared with observations could be a good way to better connect the two sections. In the original version of the manuscript, we did not pay much attention to the validation and the reasons for the model performance. We have made major changes to the manuscript to better connect the two sections by focusing on the reasons why the model mixes out the surface heterogeneity more than the observations show. Although we chose to keep the ABL validation figure in an appendix, we add a focus in the text about the model performance and the reasons why it is so difficult to capture the details of realistic surface heterogeneity with LES. Furthermore, we have added a sensitivity analysis that we had completed looking at the large-scale forcing of the LES to a supplemental material.

Secondly, we have addressed the comment about the use of a composite golden day. We have added text in both the introduction and the LIAISE data section to better justify the approach. Lastly, related to this comment, you had mentioned better explaining our method for determine the ABL vs the IBL height from both observations and the LES model. We have added text to the manuscript as well as further information in this comment to address this.

We have addressed your specific and technical comments below in more detail. Your comments are the red bullet points, and we address them in the black font. We have grouped similar comments together to address some of the comments based on theme. Finally, in some cases, we show the specific textual changes that we have implemented to the manuscript in indented italics.

**Specific comments**

- l. 4 -6: This sentence is a bit unclear/ hard to understand at first – reformulate? "strength" → "magnitude"?

Done.

Done

Done

Yes, but compared to the observations, which are at best 2D profiles with the LES, we can capture three dimensional processes, for example the secondary circulation. Regardless if she show the results in 3D or not, the processes that are influencing the results are 3D processes.

We have clarified the meaning, and the reason we stress this, in the text.

Done

According to Raasch & Harbusch (2001), the strength of the secondary circulations indeed depends on the intensity of the heterogeneity, while its structure (or presence) depends on the background wind. Although in that paper, they did not explore this parameter space as thoroughly as others. (van Heerwaarden et al., 2014) did this more explicitly. They found that in the absence of wind, even with a low intensity heterogeneity, the effects of the heterogeneity does not vanish within a day.

Yes, like Patton et al. (2005) and others found, when the scale of heterogeneity is greater than the ABL depth, we would expect that a secondary circulation is more likely to form. However, all of these experiments use more idealized wind/land surface types. Since we submitted this paper, there was a paper by Paleri et al. (2024) which looked at the influence of unstructured surface heterogeneity on the presence of secondary circulations. They found that the presence of these circulations is dependent on the effective length scale of the heterogeneities, similar to our multi-scale method. We are not aware of any other studies that looked into how secondary circulations occur with unstructured/multi-scale heterogeneity like in our case.

Lunel et al. (2024) used the idealized formulation from Segal & Arritt (1992) for the LIAISE domain, and they found that the minimum length scale of heterogeneity of 3 km, which is quite a bit larger than the length scales we found in this study. However, they used a stronger contrast between the wet and the dry landscape scales because they applied irrigation evenly over the entire irrigated region. And they did not include any field scale variation in land use and irrigation schedules. We think that the novelty of this approach is that we are using a more realistic application of irrigation and surface heterogeneity.

We find that with realistic, microscale heterogeneity, the scaling laws from previous studies do not necessarily hold.

We added parts of this text in the discussion section.

- l. 103-104: Not clear. Is the cool and moist easterly wind the sea breeze?

Yes, we clarified this point in the text.

- l. 132: The citation of Brilouet 2021 does not correspond to LIAISE data. Please refer to the right data paper.

This paper describes the SAFIRE aircraft and the applied data processing techniques of the SAFIRE data. Indeed, it does not refer directly to the LIAISE data. We changed position of the citation to clarify what the citation actually should be referring to.

- l. 136-140: This paragraph is key to the methodology and needs more detail in my opinion. How is the golden day calculated? Is it just an average of the 3 days? How are the wind fields generated? And after averaging potential temperature, specific humidity (and 3D winds?) separately, do the newly created vertical profiles still make sense physically? Is there a risk of having inconsistencies for some time step (e.g. strong negative vertical gradient of virtual potential temperature, divergence of horizontal wind speed and positive vertical speed w?)

Yes, the golden day is a mean of the three days. The wind fields are generated by averaging components (u and v) separately and computing wind speed and direction. In all cases, we do all statistical computation (e.g. turbulent flux calculation, etc.) before averaging across the days. From these results, we do not see any inconsistency between time stamps. In fact, because we smooth the signal noise, and thereby get a more temporally consistent situation than individual observations show.

Based on this comment and some of the aforementioned comments, we added additional explanations about the selection of our composite day approach to this section in Section 2.2.

Has this approach been used in other LES studies?

We are not aware of this approach being used before in LES studies. But the idea of a composite day has been used more frequently in atmospheric sciences (e.g. Grotjahn & Faure, 2008) and in boundary-layer meteorology (e.g. Angevine et al., 2001; Barrett et al., 2009; Jury et al., 2009; May & Wilczak, 1993). In these cases, they all select a number (between 10 and 50) of "similar" days based on their interest to average into a composite. In the cases of the ABL papers, they each preformed an analysis to look at the structure (e.g. winds, temperature) of the ABL on these composite days. In all of these cases, they chose to use a composite day to get physical insight about the average features of the ABL.

Sounds like it makes this study an idealised study? Does it?

We do not consider this to be an idealized case because so much of the realistic distribution of surface fluxes and large-scale forcing directly from observations. Furthermore, we nudge the domain towards observations to ensure that the state variables of the ABL are consistent with a realistic situation. We consider an idealized case to not be directly tied to a single situation/location on earth. Examples of truly idealized studies about surface heterogeneity include (Patton et al., 2005; Raasch & Harbusch, 2001; van Heerwaarden et al., 2014), where they look at how the ABL responds to idealized patterns of surface

heterogeneity. That being said, because we use a composite day, this is also not a case study LES. Rather, it represents a realistic LIAISE composite day.

Why didn't the authors use a single realistic day (which would have made it easier to compare the LES results with the observations)?

We chose not to use a single day (and instead used our composite day) because there was missing observational data over single days, and from one day alone, we would not be able to address a more typical situation in the area. We have addressed this point in the previous comments as well.

We have addressed these questions in the text:

*We have chosen to use a composite day in this experiment to maximize the data usage. Because the observations are not all continuous in time, by combining multiple days over the afternoon period, we can create a composite dataset that is more representative of the variability of the atmosphere in the afternoons. The composite day is composed of three days – 20-22 July – which are characterized by the thermal low which developed in the north-central area of the Iberian Peninsula. During these days, synonymic conditions were similar. There was relatively weak synonymic forcing. Over each of the days, there was a secondary thermal low that developed in the Ebro River Valley, and in the late afternoons, a sea breeze bringing relatively cool and moist air from the Mediterranean Sea arrived in the study domain. Furthermore, there was little daily variably in surface fluxes at each of the SEB sites over this period. We calculate the golden day by averaging the available data per 30 minute period across all days. We do this both for surface fluxes and the atmospheric fields from the radiosondes, including wind speed, specific humidity and potential temperature (mean values and daily variability shown in Fig. 2). The benefit of using a composite golden day for analysis is to gap-fill missing data, reduce the spikes in the data, and to create an atmospheric situation that is typical for the area and a thermal low day. Note that in all cases, all processing (e.g. computing turbulent fluxes) and normalization was done on individual observations before averaging to create the composite day. By using a composite day, we create a situation that is realistic for the LIAISE experiment but not a real/case study simulation. This allows us to focus on the most important processes that control land-atmosphere interactions across the spatial scales.*

- figure 2: What are the shaded areas? Standard deviations calculated over 3 days? In relation to the above comment, it would be better to plot the three radiosoundings used directly (e.g. with finer lines). Also, adding the wind speed and direction to the vertical profile would be very helpful in understanding the situation.

Yes, the standard deviations are computed over 3 day period. This has been clarified in the caption. We have selected to show the composite day because that is what the model is using and verified against. Although we lose some detail in the averaging scheme, we think it is more fair to show how the model is run and is verified than the details of each day.

- l. 147: Below 500m there appears to be a well mixed boundary layer, but above 500m it appears to be quite stable. The mixing ratio does not seem to be well mixed either. Since you mention the fact that the atmosphere is stable at 10 UTC, this just looks like a classic boundary layer evolution in a stable atmosphere. This should be discussed in the text and perhaps related to the TKE profile of Figure 3 (see also the related comment).

- l. 155: "unstable layers near the surface in the dry area" → Yes, it seems to be unstable near the surface, but it is not clear up to where it can be considered unstable. The mean profiles of potential temperature at 12 and 14 UTC show slightly stable boundary layers between 500m and 1500m, why that? Is it an effect of the three days averaging?
- l.155: "IBL in the wet area" → It is not clear why these are internal boundary layers and not just the classical ABL. This IBL could be an artefact of the averaging process. Further investigation of the characteristics of the atmosphere between 500 and 1500m would be interesting to clearly define IBL vs ABL here.
- l.162-163: Give the ABL heights found with this method for each time stamp of fig.2, maybe in a separate table or in appendix.
- l. 151: "presence of a secondary circulation" → would be interesting to have it confirmed with vertical profiles of wind, as mentioned above

In several of your comments, you mention the determination of the boundary layer height. We agree that is an important distinction to make. For our purposes, we chose to use a parcel method. In this method, base the ABL height off of the near surface potential temperature plus a given $\Delta\theta$. We have chosen the $\Delta\theta$ to be 1 K in this case. We use the parcel method because it takes into account how far an air parcel could rise before hitting a strongly stable layer. In Fig. R1, we show the individual radiosonde observations that are used to create the composite day. As you can see, there is a range within these observations. The days get warmer and drier, and the timing of the sea breeze differs slightly. We have previously mentioned the reasons why we have chosen to use a composite analysis. We have chosen not to show Fig. R1 in the paper directly because it is outside of the scope of the study.

You are correct, it does appear more stable above 500 m in the averaged figure. We have changed the text here accordingly. The stable layer above 500 m is apparent in individual profiles as well as the averaged profile. We selected to show only the average profile in the paper because it is what was used to run and to validate the model. The variability between the 3 days that make up the composite day is captured in the uncertainty range that we provide with these profiles.

In Fig. R2, we focus on only the 12 UTC time and a number of different methods for determining the ABL height. As you can see, each method yields very different ABL heights, and the heights of the relatively well-mixed specific humidity and potential temperature differ. Furthermore, the radiosondes show a presence of an internal boundary layer. Previously, we had not quantified the height of the IBL because it was so variable among the days. Now, we select the lowest stable layer in the ABL as the height of the IBL. We have added this definition to the methods section of the text.

In terms of a potential observed secondary circulation, we refer to Fig. R1. We show the wind direction from the radiosondes. If the flow was a pure secondary circulation (with no background wind), then we would expect the wind direction to be from the west at the surface in the dry, and a return flow from the east near the top of the ABL. At times 12 and 14 UTC, for some days (21-22 July), we do see a turning of the wind from westerly near the surface to easterly closer to the top of the ABL. However, because of the background winds, this signal is not very clear.

[Figure]

*Figure R1 The radiosonde profiles for each site over 20-22 July 2021. La Cendrosa is the irrigated alfalfa field and Els Plans is the rainfed fallow field.*

[Figure]

*Figure R2 The individual radiosondes over each day of the study. The points indicate different methods to determine the ABL height.*

Based on these comments, we will add an appendix with the ABL heights in table form for the composite day for both the wet and the dry radiosondes. In this table, we will calculate ABL height for each day separately and then average to get the composite case. This makes the procedure consistent with the averaging procedures used in the rest of this study. We will also add text in the manuscript to describe the indication of the IBL using observations.

- Figure 3: add in the legend what are the error bars.

Done

- l. 172: "there is a weak moisture flux near the surface" → rather seems to be virtually null

Done.

- l. 176: "strong TKE" → I would not consider a value of 0.5 m2.s-2 a "strong" TKE

The "strong TKE" was referring to the dry landscape (2 m2 s-2) not the wet landscape scale as was written. We fixed this error and modified the statement to stress that it is strong *relative* to the wet landscape.

- l.176 – 179: How do you explain having values of TKE of about 0.5 – 1 m2 s-2 in the free troposphere? The TKE values shown in Figure 3 give the impression that TKE does not vary with height. It would be interesting to show in appendix similar profiles but for the morning.

Yes, it would be interesting to similar profiles from observations in the morning, however, there were no measurements taken at the time from the aircraft. We are not aware of seeing TKE observations from the free atmosphere in literature, although modeling results do show that TKE is near-zero.

- l. 180: Why "conversely"? The text was already discussing the wet landscape before.

The previous paragraph was referring to the dry landscape. There were errors in the previous paragraph, everything should have been referring to the dry landscape. Thank you for pointing this out to us.

- l.196: Why not using TKE here, as for Fig.3? Comparison and understanding would be made easier.
- l.199: Why is the flight data is not average on 20 to 22 July like for fig. 2 and 3?

We address the two previous comments here.

TKE requires an average in time/space. As this is aircraft data, we could in effect get one value for each the wet and the dry landscapes. We could show a sort of "instantaneous TKE" to show spatial variability. We chose to show this quantity over "instantaneous TKE" because by taking the square root, the differences are evident while preserving the details in both the wet and the dry scale. Plotting this as TKE exaggerates the dry compared to the wet, so the wet looks like 0 while the dry dominates the scale. This way, the main point that the dry is much stronger than the wet is still evident, but we can also see the details of the signal in that the local peaks related to the temperature ramps. We included the average TKE at the top for comparison to the Fig. 2.

Unlike Fig. 3 where we showed the composite of the fluxes taken over the course of the afternoon with a convective BL, in Fig. 4, we show one day. The composite figure would still show the increase of TKE and temperature variance in the dry landscape compared to the wet, but because we show a turbulent signal in Fig. 4, instead of statistical signal (like Figs. 2-3), the details would be smeared out. We want to make the point here about the suspected presence of thermals and are more coherent ABL in the dry than the wet landscape.

- Figure 4: y-axis is uTKE and but mean values on the top left corner are TKE… I think using only TKE here would be better. Also add the altitude a.g.l. or a.s.l. of this flight.

We above explained our choice of using the square root of an instantaneous TKE, and we added the flight altitude to the caption.

- l.215: "superadiabatic lapse rates in the lowest hundreds of meters" → Fig. 2 shows superadiabatic lapse rates closer to the surface in my opinion

We changed this to state … "just above the surface".

- l.220: "evident" → do not agree, should be discussed more in a more balanced way, to relate with previous comments

This has been fixed there.

- l.253: Add a few sentences or a paragraph to resume the results of the comparison between the LES and the LIAISE data (cf also comment on Appendix A below). Specifically you should mention the shortcomings of the LES outputs and justify why you don't observations anymore in this section.

This is a good point. We added that at the end of the model configuration section. We have also added an appendix (now appendix B) on the sensitivity studies that were done to reach this case.

*In Appendix A, we show the validation of the LES based on the composite golden day LIAISE profiles from the radiosondes (Fig. 2). We see that, although the model is constrained into the correct order of magnitude as the radiosonde observations, the difference between the wet and the dry local scale is smaller in the model than in the observations. This is likely because there*

*is too much mixing in LES. We performed a number of sensitivity studies (Supplemental Material) where we tested the influence of large-scale forcing terms on the model results. We used a suite of forcing terms based on the ERA5 reanalysis and observational data. The geostrophic wind is too high in the ERA5 which cause a high bias in wind speed in the model compared with observations. Therefore, we opted to run the model entirely forced by observations. Furthermore, from our sensitivity study, we see that even ERA5 does not capture well the larger scale situation in the LIAISE domain, no matter which day we chose to run. This further justifies the use of a golden day approach. We know that the local SEB's are relatively similar across days, so all ABL variability would have to come from the larger scales. The large scale forcings, however, exhibit large uncertainties and therefore, we cannot capture the details of each day's ABL. A composite day is then the best approach to focus on the influence of irrigation-induced heterogeneity on the ABL.*

- l. 259: The most energetic eddies of the turbulence are smaller in the surface layer and in the entrainment zone. A vertical resolution of 25m means that the eddies close to the surface are not explicit in the model, and therefore the LES is not an LES in the near-surface troposphere. In the same way, eddies in the entrainment zone are smaller than in the well-mixed part, and your model may not be a LES in the entrainment zone. Do you have an idea at which heights your model can be considered an LES? As the IBL above the alfalfa local scale is rather low, can the model be considered an LES in this part of the atmosphere? Please quantify how much of the TKE is explicitly resolved in your LES at different height. This should be done ideally for the wet and dry landscape.
- l. 382: In the surface layer, with your resolution, the model is very likely not an LES… How is your TKE computed here? Is it mainly a sub-grid TKE here?
- l. 383: Similarly in the entrainment zone, the model is likely not an LES either

The model includes a subgrid scale parameterization of turbulence. At the surface, the entire flux is from the subgrid cell parameterization, and by the third grid cell ($z = 75$ m), the subgrid flux accounts for less than 5% of the total flux. Above 150 m, the over 99% of the total flux is resolved. In the entrainment zone, the SGS flux becomes relatively more important, mainly because of the relatively small total flux. In the entrainment zone, the unresolved flux accounts for 20% of the total flux. This peaks above the ABL height (at $z/z_i = 1.2$).

We have added text to the manuscript to say this more clearly in Section 4.1:

*… Near the surface, a component the turbulent flux comes from the subgrid scale parameterization. This is done with an eddy diffusivity closure with the Smagorinsky sub-grid-scale parametrization (van Heerwaarden et al, 2017). The parameterized flux accounted up to 5% of the total flux from the surface to 75 m, and it fell below 1% above 150 m. Therefore, we consider the resolved fluxes to be sufficient above this level. In the entrainment zone, the parameterized fluxes become important again, accounting for 15% of the total flux. With this horizontal resolution, we can capture the presence of individual fields.*

[Figure]

*Figure R3 The regional scale resolved and subgrid scale fluxes at 12 UTC.*

- l. 259: What do you call equilibrium with ABL in the context of a heterogeneous terrain? Do you have a reference for justifying it is large enough?

I agree that equilibrium is probably not the best word here. We mean that when you have periodic boundary conditions in an LES, the turbulence that exits one side of the domain reenters on the other side. Because the dry landscape is only 35% of the land area in the regional scale, we wondered if we ran only the LIAISE regional scale, the entire domain would look more like the wet landscape than the dry landscape. We tested this in a domain size sensitivity study, and we found no difference in statistics at the regional and landscape scales when running the extent of only the regional scale or when running it with a buffer zone of 10 km in all directions.

For this reason, we expect that this buffer zone is not so important to have in a case this size, but to be safe, we added one to make sure the air entering the LIAISE regional scale is realistic to the air mass in reality. From the aircraft observations, we find an integral length scale of the turbulence near the top of the ABL to be up to 300 m. Velocity fields are on the same order of magnitude. And in our LES, we see the integral length scale of w is 800 m. Based on that, we expect the turbulence to adjust to the land surface instead of being a function of the periodic boundary conditions.

[Figure]

*Figure R4 Integral length scales of potential temperature and specific humidity for the aircraft legs over the wet (green) and dry (yellow) landscapes over different days (symbols).*

We will change the text to clarify what we mean by "equilibrium".

- l. ~264: Do you consider orography or a flat terrain? In the latter case, a map of orography could be added in Fig. 1. Also if orography is considered, mention the limitations inherent to the fact of using a single radiosounding to represent all the domain in such a complex terrain.

We do not take orography explicitly into account in our simulations. The simulated area is flat terrain, which is not entirely realistic, but is a good approximation because we are simulating a closed valley. Ideally, we would use large scale forcing from a global model like ERA5 so that the topographic effects are explicitly included, but ERA5 didn't capture these effects well, so their inclusion is through the observations alone.

- l. 264: refer to comment on l. 136-140, and recall the limitations of such an approach based on an average day.

We have added a longer discussion in the methods section, so we will not repeat that here. Please refer the comment above for our response to this point.

- l. 272: If I understand well, the same advection is considered at all heights of the domain, based on the advection measured near the surface? Please discuss the limitations of such an approach.

Yes, this is correct. The advection terms are calculated from surface observations taken at 10 m above the ground. The primary limitation is that it assumes that the advection is constant with height within the ABL. While this is likely not true, as seen in the complexity of the individual radiosondes (Fig. R1), it yielded the best results for the LES overall (see Appendix B in paper). Our approximated advection term yielded better results than the ERA5 advection term. Also, by following the approach we are consistent with a previous study we did in the region with a conceptual model where we also estimated the advection terms from observations as ERA5 is not performing well in this domain (Mangan et al, 2023).

- l. 278: How the roughness lengths are determined in Mangan et al 2023a? I didn't find much information on that in the article.

The roughness lengths were approximated based on the vegetation height. We assumed that the roughness length for momentum is 10% of the vegetation height, which was measured directly during the experiment. We assumed that the roughness length of heat is 10% of the roughness length of heat.

This is clarified in the text at that location.

- l. 291: The choice of the parcel method should be more discussed, with the limitations and advantage of this method (e.g. doi.org/10.1016/S1352-2310(99)00349-0). I guess this method is particularly relevant for assessing the possible ABL top from a single field, and therefore to study the local scale here. However the actual modelled ABL top is also influenced by larger scales as you show later, and therefore other methods for the ABL heights could be used to make the most of the LES, specifically for the landscape and regional scales (e.g. gradient of TKE, of potential temperature, etc).
- l. 297: You consider the ABL height of the wet and dry landscape as the mean of the local scales included in each landscape if I understand well. Could you measure the height of the ABL with a different method more adapted to the landscape scale and check if you find the same value approximately?

We will address the previous two comments together here.

We did chose this method precisely for this reason. It matched the method we used to define the ABL height from the radiosondes. Philibert et al.(2024) did an analysis of the different ABL height methods from the LIAISE experiment using a number of observations. We computed the same metrics for ABL height for the LES and found that the parcel method was the most consistent across time and best matched with the observed profiles. For those reasons, we selected the parcel method.

Furthermore, we compared the ABL height calculated for each grid cell averaged over the regional scale with the ABL height computed over the spatially averaged potential temperature profile. This is a proxy for a non-local calculation of ABL height. We found that the heights are nearly identical ($r2 > 0.99$ and $p = 9.1e{-}29$).

We added some clarification about this point in the text.

- l. 300: This value of specific humidity is very low, are you sure the unit is the right one?

It is the advection of specific humidity, not the actual value of specific humidity. This is the correct unit (g kg-1 hr-1) and value based on our advection estimation described in Mangan et al. (2023). The impact of the temperature advection on the ABL height is larger than that for moisture. This was discussed in Mangan, Hartogensis, van Heerwaarden, et al.(2023) in the context of surface fluxes.

- Figure 7: What is the shade around the lines representing local scales (alfalfa and fallow)? If it is a standard deviation how do you calculate it? Also add to the legend that the standard deviation correspond to the shade.

Yes, the shading around the lines for local scales is also standard deviation. Like the other scales, it is computed spatially at a single time. While the local scales are single fields, there are several grid points per field. We located the locations of the specific fields using lat/lon locations and ensured the correct land use type for these fields. At the 30 m resolution, there are approximately 1000 grid cells in the alfalfa field and the fallow field, which is not enough to get robust estimates for the average/standard deviation and hence, the results at this scale are noisier than the larger scales.

We will add this more clearly to the text and in the caption labels.

- L. 290-300: each time you refer to Fig. 7 you actually refer to Fig. 7 (a). Please be more precise in the referencing of the figures by adding the letter when appropriate.

This has been clarified in the text.

- l. 301: You didn't comment the Figure 7 (b). Remove it or discuss it. Specifically it is interesting that you find EF > 1 for three scales including the largest one. It seems rather early for Spain.

This has been clarified in the text.

- Figure 8: Why is fallow scale is not shown for 08:00 UTC?

We are checking this out.

- l. 305: "and z/zi ∼ 0.5 at all times of the day" I don't understand this part.

That  has been fixed to read:

*The fallow local scale has the highest heat flux from the surface up to  $z/z\_i$ ~0.5 at all times of the day.*

- l. 309: "the increase OF heat flux"?

Done.

- l. 310: z/zi ∼ 0.2

Yes, that has been fixed.

- l. 310: What are entrainment fluxes? Fluxes in the entrainment zone? Then it is even way above 0.5.

Yes, entrainment fluxes are fluxes from the entrainment zone into the ABL. We do not necessarily mean the flux interface between the entrainment zone and the ABL.

- l. 317: High moisture fluxes over the dry landscape are also found below the entrainment zone, so the entrainment of free troposphere dry air may not be able to explain such high moisture fluxes… Also it is not only the strength of the entrainment that determine the moisture flux but also the gradient of humidity between the ABL and the free troposphere. And in the case of the dry landscape I would expect a relatively weak gradient in comparison to the wet landscape. Could it be the signal of a secondary circulation that you develop afterward?

Yes, that is a good point. We also see this in the radiosondes that there is an increase in concentration in specific humidity at the top of the ABL in the dry landscape, which we attribute to advection. In a later study (out of scope for this paper), we did a more thorough budget analysis of the moisture. We found that over the dry landscape, the term $\overline{w}\frac{\partial q}{\partial z}$ integrated over the entrainment zone was negative in the dry landscape and positive in the wet landscape.  We attributed this to the vertical velocity more than the gradient in q at the top of the ABL.

We added this potential interpretation to the text.

- l. 326: Fig. 5?

We do mean Fig. 4 because it is the same transect that the air plane flew. We clarified this in the text.

- l. 338: Can you please develop the physical meaning of your characteristic length scale?

This has been added to the text. We have added some sentences to explain that the characteristic length scale is the length scale that is most important for describing the spatial variability of the physical features. We also stressed that these characteristics length scales can be compared with the length scales of heterogeneity that we use in this analysis.

- Figure 9: At what time are the spectra calculated?
- Figure 9 a, b, c, d: Add to the legend what is the gray dashed line. For panel d, the spectrum of the vertical velocity is expected to follow a law in k-2/3 of the inertial subrange of the turbulence. Can you please represent such a law on this panel (e.g. https://doi.org/10.1007/BF00122327) and discuss it to assess the behaviour of the turbulence in your LES?Figure 9 e: change y-scale to better see the low values of the integral length scale.

We will answer all of your questions about Fig. 9 here. The spectra is calculated at 12 UTC. We have updated the legend of Fig. 9 to add this information.

We show an updated version of Fig. 9 below to address your other comments. We have added the k-2/3 line to the figure in the inertial subrange. Furthermore, we extended the range of We could do that, but we chose not to as in the layout of the figure, the axis is dominated by the length scale of w. It is not be possible to see any difference between the H/LE and zi in this case.

[Figure]

- l. 340: "1111m at 12 UTC" for which area? The regional domain? If it computed over the regional domain, then the zi considered is the same everywhere, and you could add that it is at 555m agl. It would make more sense to relate it to the landscape scale zi, and in this case you should give the zi values for each landscape.

That is correct. This has been added to the text.

- l. 341: "the gray DASHED (?) line" If you indeed refer to the gray dashed line, please add it to the legend of figure 9.

This has been added to the caption of Fig. 9.

- l. 344: It is indeed likely that the microscale peak is linked with the microscale of H and LE, but this very likely due to the parcel method that uses the near-surface temperature to compute the ABL height. I think it just shows that the parcel method gives ABL height that strongly depend on the field scales. If you do the spectra of ABL height calculated with a different method not so dependent on the surface, would this peak still appear?

  Then, the upturn for the high wavenumbers of a LES can also be due to numerical filtering issues (for ex. cf fig 10 of doi.org/10.1175/MWR2830.1), how to be sure this microscale peak is not due to that?

If I understand your question and Fig. 10 correctly, you are asking how we know that the microscale peak is not due to aliasing? We see this microscale peak in both the LE and H spectra and the $z_i$ spectra. In all spectra, our smallest wave length is 2*grid spacing (=60m). Spectra computed from H and LE are an input into the model from observations and are a direct function of land use, and therefore, should not be impacted at all by the model filter as argued by Skamarock, 2004.

Regarding the spectra for $z_i$, there are two potential issues you list: (1) model filtering effects , and (2) the local computation of the ABL height. (Otte & Wyngaard, 2001) found that with model resolutions up to 15 m, there is an unphysical peak in the velocity spectrum. They used the subgrid sale model from Deardroff (1980). When they tested this with a nonlinear SGS model (e.g. allowing backscatter from parameterized scales to resolved scales), this problem was minimized. Our LES, MicroHH, uses a filtering length scale as defined by Mason and Thompson (1992) to allow a backscatter of energy from the subgrid scale to the resolved scale, so we don't expect aliasing based on the model filter. Furthermore, we don't see this in the w spectra (as it was seen in Otte & Wyngaard, 2001), so likely aliasing is not the cause of the microscale peak.

The second point, about the local calculation of ABL height is more likely to be the cause of the microscale peak in $z_i$. The near surface temperature responds faster to surface fluxes, while the temperature near the top of the ABL respond likely to a larger area. We tested a number of different methods to compute ABL height, including a Richardson number method and maximum/minimum gradients in potential temperature and specific humidity. We found that the most consistent results compared to visual inspection of profiles matched with the parcel method.

- l. 349: "IS" more classical? What is the classical spectra of spatial surface fluxes? It does not necessarily follow a law in k-2/3

This has been fixed.

I agree with you that we would not expect the k-2/3 law  because it is a function of land use type and not turbulence. While I had expected the spectra of $z_i$ to look like that of w, this was not the case. Also, I am not aware of a reported $z_i$ spectrum in the literature, so we removed the first part of the sentence.

- l. 358: "lessons the length scale"?

Thanks for pointing that out, and we changed this to "reduces".

- l. 359-360: Why the length scale of vertical velocity does not decrease abruptly in response to the sea-breeze like for the length scale of $z_i$?

We think that it has to do with the fact that the spectrum of w is computed at a fixed location within the ABL. In Fig. 7, we see that when the sea breeze arrives, the ABL collapses, but the evaporative fraction

is still less than 1 for most scales, meaning that there is still buoyancy flux at the surface that maintains the variance of the vertical velocity.

- Figure 10 a,b,c: Place the alfalfa and fallow field on the x-axis. Add in the legend that the transect corresponds to the box of Fig 6 c. The distance 0 does correspond to the border between wet and dry landscape from figure 6, why is that? What is the white line in panels a, b, c? More generally for figures, better describe all features of the figure (colors, lines, etc) in the legend, and no need to repeat it in the text then.

The caption has been rewritten to better include these comments.

- Figure 10: in legend: "dashed lines" → "dotted lines"

Fixed

- Figure 10 d,e,f: The fluxes in the entrainment zone are not readable. Put it aside in a different panel or remove it.

We will remove the entrainment fluxes from that figure.

- l. 365: Fig 6a → Fig 6c

Fixed.

- l. 366: "wind direction is predominantly from the west" This wind direction should have been described earlier when discussing the weather situation on the golden days (l.136-140).
- More specifically one could expect from a thermal low situation on the iberian Peninsula to have a eastern wind in the Ebro basin. Why is the wind westerly here?

We address the two previous comments below.

The winds are primarily from the West-Northwest before the sea-breeze arrives at the end of the afternoon. The (relatively weak) geostrophic wind is westerly. There are two thermal lows during these days: one in the center of the Iberian Peninsula, and one in the Ebro River Valley (see (Jiménez et al., 2023)). The presence of the thermal low in effect weakens the synoptic scale geostrophic wind but doesn't remove it all together. So the surface winds are mainly from the west.

As recommended, these specifications were added to the description of the golden day.

- l. 367 to 371: You describe the features of the figure here. This should be done in the legend, and the text should be used to interpret what is shown in the figure.

This has been changed in the text.

- l. 372: "increase in zi": How do you see that? Is the white line the ABL top? If so, I would expect from Fig 6c that the ABL goes from about 500m to 1500m.

Yes, from the white line. This is clarified in the next.

- l. 376: It's tricky to compare the measurements with the LES in view of the comparison results shown in Appendix A... It should at least be pointed out that the LES results are not directly comparable with the measurements.
- l. 458: I don't really agree on the fact that it is a realistic case. It mixes weather situations of three days without proving that it correspond to a case that happened in reality. Moreover

this methodology makes it hard to compare the LES results to the observations. The only part discussing this is put in appendix, and the results shown in appendix are not convincing.
- Appendix A: the cool IBL observed in the 500m agl is not reproduced by the LES, even at the alfalfa local scale. Why is that? More generally, I miss a critical description and interpretation of the LES results. Or maybe stress that the LES results should be considered as an idealized atmospheric setup but with a realistic surface flux map, and therefore that the comparison to the observation is not useful.

There is very few comparison between the LES and the observations along the article. I guess the reason for this lies in this appendix. Maybe this figure deserves its place in the body of the article?

In a comment above, we have addressed already why this is not an idealized simulation and that you are correct that is not a real or case-study type of simulation (e.g. Schalkwijk et al., 2015; van Stratum et al., 2023).

However, we disagree that they are not entirely comparable. You mention that because the LES shows too much mixing between the wet and the dry landscapes, and it doesn't capture the layering in the ABL over the wet landscape, that the observations are not comparable. It is correct that these features are missing, and because of that, we can learn more about the performance of LES in heterogeneous landscapes.

In literature, there are few realistic LES cases with surface heterogeneity. Again, by realistic, we mean that the initial atmospheric conditions and land surface cover are based on observations. Not necessarily that the cases can be used for forecasting purposes. In this manuscript, we cite Maronga & Raasch (2013) where they do run the LES and validate it with radiosondes for a single day in the experiment. Based on their experimental design, they change the geostrophic wind and domain size and evaluate how that influences the ABL. They find biases up to 1 K and 0.5 g kg-1 in the mixed layer. They compared domain spatial averaging of the LES with a single observation, so we cannot say how the individual profiles look in the LES. The LES also doesn't capture all of the complexities of the radiosonde observations including the layering in the radiosonde observations.

More recently, there were realistic, nested LES studies reported over heterogeneous land surfaces for the CHEESEHEAD experiment. Paleri et al. (2023) introduces the case, and their verification with radiosondes have biases on a similar order of magnitude as Maronga & Raasch (2013). The bias in potential temperature is approximately 1-2 K during well mixed conditions and specific humidity between 1-2 g kg-1. Like Maronga & Raasch (2013), they validate only with one radiosonde. They don't validate on radiosondes that show the atmospheric heterogeneity. Our results are in the same order of magnitude as these studies.

Furthermore, to capture heterogeneous cases in LES, capturing the large-scale forcing is extremely important to get right. If mesoscale forcing is too strong, then the influence of heterogeneity quickly mixes out. Like cite Maronga & Raasch (2013), we preformed multiple sensitivity studies to the large-scale forcing components including geostrophic wind and advection of scalars and the wind. We have added a summary of these experiments to a supplemental material. What we find is that the wind and the geostrophic wind are the most important features to get correct to maintain the observed heterogeneity. One issue we ran into during this study was that large-scale models, like ERA5, do not well capture the mesoscale processes in the area. For that reason, we opted to run entirely forced on

observations so that we could control the case to the best of our knowledge. Although we hypothesize that our missing process in the LES comes from mesoscale processes, based on the current knowledge in the area, we do not know yet what process we are missing and why the global scale models preform so badly here.

Although fixing the wind lets us have an observable difference in scalars among the scales, the differences in the radiosonde observations are more pronounced. Another reason for this could be something within the LES numerics that causes too much mixing. This would also relate to how we could not capture the distinct layering in the potential temperature profiles at the wettest fields in the wet landscape. For example, we impose a constant initial surface pressure level throughout the domain, which might suppress the formation of pressure-based flows. In our case, there is topography that could also play a role. From discussions with others who do LES in realistic, heterogeneous domains, this "too much mixing" problem is relatively common.

Overall, it is very difficult to capture these features in the LES. We find that the results are most sensitive to large-scale forcing including geostrophic wind and advection. Even with our case where we think we have reasonable land surface fluxes, we cannot fully capture the details in the ABL, which tells us that we must be missing a process. We do think this is a reasonable case because we capture the bulk characteristics of the composite day (Appendix A).

Based on your comments, we will add a short section in the discussion section to discuss the validation of the model and our current challenges/drawbacks of our LES experiment.

- l. ~371: It is not said how TKE from the LES is calculated. From the figure 10, we can have the feeling that the small whirl at distance=800m and z=700m should be considered as turbulence more than mean flow. More detail should be given on the separation between mean flow and TKE in the LES. Also do the values of TKE include the sub-grid TKE or not?

In this case, the TKE was computed using the variance of each transect for each time stamp. We first isolated the aircraft transect, then computed the mean and fluctuations of the velocity field (via Reynold's averaging) across the transect. Finally, we computed TKE with the fluctuations of the velocity field. Like we did for the aircraft, we used spatial averages, not temporal in the LES. So the transects are instantaneous in time. We did not use any filtering method to remove the mesoscale.

- l. 385: Clarify the nature of the counter gradient and what it means.

We added a short sentence to describe the "along-gradient" fluxes based on Fig. A1, to clarify what we mean. Therefore, the section now reads:

> ...For both the heat and moisture flux, there is a counter-gradient flux in the dry landscape. Based on profiles of potential temperature specific humidity from the LES (Fig. A1), the along-gradient flux in the entrainment zone is warm and dry air brought from the free atmosphere into the ABL ($\overline{w^{\prime}\theta^{\prime}} < 0$ and $\overline{w^{\prime}q^{\prime}} > 0$). The counter-gradient fluxes could indicate the influence of the wet landscape on the ABL in the dry landscape. For TKE, there are pockets of strong TKE in the entrainment zone in the wet, although there is a skew to the higher TKE in the dry landscapes.

- l. 389: What is the reason for this small circulation? Or could it be a large eddy of the CBL? In the latter case, it could be considered as one of the large eddy of the turbulence more than as a secondary circulation. More details must be given on what is considered turbulence and what is mean flow in Figure 10.

If I understand your question, you are asking why we say that this is a secondary circulation (where it would persist over hours) instead of turbulent signal? We did not do any mesoscale filtering in our analysis. We do subtract the mean wind from the signal. It is possible that this is a turbulent signal, and we are checking in other times to see the persistence of this feature.

We will update the text with further description of its persistence and if it is truly be attributed to (as we suggest) a circulation/persistent feature.

- l. 390: "One a stronger secondary circulation" I don't understand this sentence.

We have rephrased this sentence to correct the typo.

- l. 392: Interpreting something on IBL from the absence of IBL signal in the figure 10 seems problematic…

We have rephrased to make more clear. We don't observe an IBL at the landscape scale, which in itself is the result and interpretation.

- l. 410: Specify the unit of the blending height and meaning of it different values. As I understand the unit is [1] and a values of 1 means that the blending height is only found at the ABL top. Is that right?

By our definition, blending height is unitless. We defined it as a binary function – 1 meaning it is blended and 0 meaning it is unblended. In line 410, we are just defining the general meaning of blending height.

We have clarified this point in the text.

- l. 411: $C_{v,b}$ is the field averaged at the surface then?

It could be. We just want to make clear that this is a general definition of blending height. The actual blending zone can be determined in a number of different ways. In the paper, we looked at $C\_vb$ as one scale while $C\_va$ is another scale. However, we did look at the case you describe where we took $C\_vb$ as the surface parameter. We described it briefly in this paper.

Based on your comments, we realized that our formulation of the blending height was not clear. We have re-written parts of the text to clarify the interpretation of the equation and how we apply it in our case. .

- l. 421: "we analyze at the heights in which the local" to reformulate

This has been fixed.

- l. 424: "In Fig. 11, we show…" the general description of figure 11 comes after the specific description of each panel. To reverse.

This has been fixed.

- Figure 11: Change style for markers between variables. When dots overly, it is not possible to see the one in the background. Also change increase y-scale for panels c and d, points around zi are hidden.

Yes, this can be revised in the revised version of the manuscript.

- l. 449-450: "This is because entrainment introduces air masses of different properties into the ABL" This sentence is not clear. The entrained air is supposedly the same in the wet or dry landscape. As I understand it it is more the entrainment rate which modify more or less the ABL depending on the landscape scale.

I understand what you are saying, and I think you are correct. The meaning of this sentence is that the entrainment itself is not uniform. Although we expect the same air mass to be in the free atmosphere, the locations and strength of entrainment is not the same across the scales. This is what causes the atmospheric heterogeneity in the ABL.

This has been edited in the text.

- l. 462: What is the "land-surface scale"? Do you mean "land-surface" only?

Yes, fixed.

- l. 467: "evident": avoid this type of subjective judgement. Moreover I would expect negative surface heat fluxes for a stable layer… this should be discussed in the previous 4.2.1 section. Do not add interpretation of the results in the Discussion section.

This point has been fixed.

- l. 469: "Do not cross defining the landscape scale" Unclear, reformulate.

Clarified.

- l. 478: You mention a secondary circulation for the dry landscape when you discuss Figure 10. Must be clarified.

Yes, that is true. Here we mean to say that within each landscape scale, secondary circulations form. There is evidence that there is some circulation between them. We clarified in the text to be more consistent.

- l. 486 "greater than 2.5 ms−1 reduces the formation of secondary circulations if is not normal to the boundary" This should be discussed in the literature review section. In the introduction you mention that 5ms−1 of background wind is needed to prevent secondary circulations. How this threshold of 2.5ms-1 relates to the the other threshold of 5-7 ms-1?

If I understand the literature correctly, the difference is that the 5-7 ms-1 refers to the wind speed that will destroy a secondary circulation even if the wind direction is favorable for its formation (i.e. normal to the boundary). The 2.5 ms-1 is the wind speed that would destroy their formation if the wind direction is not favorable (i.e. parallel or something in between parallel-normal like we have here).

We clarified these numbers in the introduction.

- l. 506: "to represent the LIAISE." The LIAISE campaign? To reformulate.

Fixed

- l. 507: You use a realistic averaged surface flux map, but since atmosphere and surface are not coupled and the turbulence in the surface layer is not explicitly resolved, I don't think you can say that you "explicitly resolve the surface in the LES".

We fixed it to say that the surface is prescribed from observations.

- l. 520: Didn't you say the opposite at line 474 (and in the corresponding previous section)?

Yes, thank you for catching that. Indeed, it should be written as the field scales within the landscape scales are more variable than the landscape scales themselves. We fixed this in the text.

- l. 527-528: missing word?

Yes, fixed.

- l. 535: The stable surface layer is not that clear either from observation or model. To discuss in the previous corresponding section. Characterize objectively the stability of this surface layer (e.g. positive gradient of potential temperature, Richarson number, or other)

The negative sensible heat flux at the surface (Fig. 3) implies a stable layer. We will add an appendix with ABL/IBL heights for times based on the observations. In this table, we can also add an indication of the near-surface stability.

- l. 541: "The regional ABL characteristics fall between the extremes of the two landscape scales" When you discuss it (Fig 7 and 8) it feels like you just average the features (profiles, ABL top) and don't use independent measures. This sounds trivial if you just average the two landscape scales.

The mean for each scale is calculated separately. If the regional scale average is the (weighted) average of the wet and dry landscape scales, then it would then apply these are linearly processes and it would be fair for large scale models to treat these processes as such.

- l. 550: Concerning this second unblended layer due to entrainment, it is due to the fact that ABL is never blended for vertical velocity in the ABL (to deplace)

Yes, that is likely correct. Because thermals form preferentially over the dry landscape (Fig. 10) and the ABL is higher there, there are areas the entrainment fluxes are higher. Entrainment of warm/dry air (and the resulting fluxes) occurring heterogeneously is the cause of the second unblended layer.

We have rewritten the section on blending height to hopefully clarify this point and similar ones others brought up by other reviewers.

- l. 554: "bi-directional land-atmosphere interactions" The word bi-directionnal gives the feeling you studied this bi-directionnal interaction, whereas it is uni-directionnal in your case as I understand it.

Yes this is understandable. You're correct that this is a uni-directional case, but we want to make clear that this is the step towards the bi-directional case. I don't think it is fair to do this analysis of the ABL with a fully coupled land surface because that only increases the degrees of freedom. There are more tuning parameters there, so it is a step towards that. We are isolating one component, namely can we reproduce the ABL. Coupling the land surface is a next step. We rephrased the sentence to clarify what we meant.

- l. 557: "a maximum background wind of 2 ms−1" if the maximum wind is so low it should allow for a secondary circulation to develop. I think it is more an issue of wind direction. The circulation expected would go from wet to dry landscape, and it is already the direction of the background wind, so the circulation may just be included in the mean westerly wind. For the return flow of the circulation, it is indeed not visible, but it is known that this features is not systematically observed. Even though I am not an expert, I think these resources may give some clues on that: doi.org/10.1175/1520-0450(1994)033<1323:TIOLSW>2.0.CO;2 and doi.org/10.1007/s10546-010-9517-9

As stated in the previous comments, with that wind speed, even with an unfavorable wind direction, one could potentially form.

- l. 559: "combining observations with an LES case study" The article gives the feeling the observations and the LES are treated very separately, and indeed it seems it is needed since the LES does not capture all the observed ABL dynamics (Appendix A).

While it doesn't capture all of the observed ABL dynamics, I think that it captures the most important ones. Because it is forced by observations and run with realistic forcing, to the best of our knowledge, this is how the local forcing would influence the ABL. That being said, there are some processes missing that the observations show and the model is unable to capture. This issue was addressed in more detail in a previous point.

- l. 559: "increased buoyancy flux from the dry landscape influences the state variables" can you recall how?

That sentence was originally referring to the previous one about secondary circulation. The claim is that the secondary circulation alters the entrainment fluxes (shown in Fig. 10). However, this is not very clear, so we changed the final sentence all together to better explain an important point of the paper:

*... Finally, by combining observations with an LES case study, we can test how well we can capture the observed ABL in a heterogeneous region. We found that properly constraining the mesoscale forcing is vital for determining the influence of surface heterogeneity on the ABL.*

- l. 573: Please develop the surface representation issue and the assumptions you mention.

The surface representation issue mainly refers to the production of the surface flux product. These are not direct observations of fluxes from each field. Instead we make the assumption that all fields of the same crop type have identical fluxes in the region. This is likely not realistic because surface fluxes respond to atmospheric heterogeneity. We would need a coupled land surface model to the LES to address this issue.

- Figure A1: What does the shading correspond to? Would it be possible to increase the size of the figure to be able to see the details? By rotating it if necessary.

The shading corresponds to the min/max of the daily radiosondes binned by altitude. This has been added to the caption. We also enlarged and rotated the figure so that it takes up a full page.

- Appendix B: According to the shaded areas, there is very few area where the atmosphere is blended, even above the ABL. This result is surprising and may call into question your methodology for the blending height calculation. Explain why you find such behaviour above the ABL.

We find this behavior above the ABL because the fluxes themselves are small and the variation of fluxes in space are also small, so the coefficients of variations are also small. This makes the method not so

robust above the ABL in terms of fluxes. We think it is best to keep the figure as is to show potential limitations of the technique, but we have added text to help with this interpretation in Section 4.2.2.

**Technical corrections**

Thank you for pointing these out to us. We have implemented all of the corrections and textual clarifications for the points listed in your comment.

This manuscript combines surface and radiosonde observations of the atmospheric boundary layer (ABL) with large eddy simulation (LES) experiments to investigate the role of surface heterogeneity on the dynamics and thermodynamics of the ABL at a variety of spatial scales. The topic is interesting and timely for the ABL community, and the manuscript presents some interesting results, particularly on the interplay of advection and entrainment processes in shaping the ABL. However, I find the manuscript to be poorly structured and poorly written to the extent it makes it difficult for the reader to follow (some suggestions below). I highly recommend a major overhaul to the structure/writing. I also have some technical comments that I am hoping would help improve the manuscript.

Thank you for your review and your suggestions on how to improve this manuscript. In general, we understood the issue to be with the structure instead of the scientific content. We understand that the structure is unusual, and we had hoped that our reasons for choosing it were clear. Based on your comments and the comments of the other reviewers, we have made major changes to the manuscript to improve the clarify. Because the other two reviewers did not agree with your recommendations to overhaul the paper, we kept the structure as is. That being said, we did make major changes to the text to address your concerns. The main changes that we had made can be summarized as:

1. **Introduction:** In the final paragraphs of the introduction, we improved the description of the structure of the paper. We wrote more clearly where information can be found and what can be expected. The edited text reads:

   *The purpose of this study is to evaluate the impacts of surface heterogeneity on the ABL across three scales of heterogeneity over a representative LIAISE day both from a data-driven approach with the comprehensive LIAISE field campaign and modeling-driven approach with a high resolution LES. In this way, we can evaluate both the physical nature of how the surface heterogeneity impacts the ABL and the potential impacts of how resolved turbulence blends the heterogeneity with height in the atmosphere. Our approach combines observational data with an LES experiment. First, we explore the LIAISE ABL using observations. We study boundary layer development spatially and temporally by combining a network of surface energy balance stations, radiosondes and aircraft data (Section \ref{sec_2}). Based on the results from Section \ref{sec_2}, we will define more explicit sub-research questions to be explored with LES (Section \ref{sec_3}). In Section \ref{sec_4}, we run an LES experiment inspired by the observations of the LIAISE composite day to study how the observed ABLs form across different spatial scales. For the LES we prescribe the land surface with observations so that we can capture the realistic, unstructured surface heterogeneity that was observed during the LIAISE experiment. Finally, in the discussion section (Section \ref{sec_5}) we will bring model and data results together to answer the question of how the development of the ABLs differ across spatial scales of heterogeneity in the LIAISE experiment. In particular, using the LES, we investigate the characteristic length scales of heterogeneity that propagate into the ABL, and we discuss how the scales of surface heterogeneity blend in the ABL.*

2. **Justification of the composite day:** Reviewer 1 suggested that the sections would better fit together if we pay more attention to both the justification of the LIAISE golden day (hereafter called the composite day) and if we can discuss the model performance and validation better. We added text in the section "LIAISE data" to justify our approach of using a composite day, which we then refer to again in Section 3.

Furthermore, we have added the text in the section 4.1 Model Configuration to discuss the validation of the LES. We think that by discussing the validation helps to better connect Sections 2 and 4 in the text. We refer to our response to the reviewer 1 for the full textual changes.

3. **Discussion of the model validation:** Another reviewer mentioned that the two sections do not fit well together because there is not much validation of the LES. To address that, we have added text in the discussion section and the methods of the LES. We think that in this way, the observed data and the LES study are better connected.

4. **Improved description/interpretation of the blending height**: From your comments below and the comments from Reviewers 1 and 3, we have made major changes to the section. First, we improved the equations and their description of the blending height. While the methodology/equations did not change, we rewrote them to help with the interpretation. Secondly, based on your comments, we also improved the interpretation and description of the physical processes in this section. We refer to your specific comment below for these changes.

We have addressed your specific comments below. Your comments are in the red, our responses are in the black, and proposed manuscript changes are in the black italics.

Specific Comments:

1) Figure 2: The authors correctly discuss that the upper part of the ABL above the wet surface (Alfalfa) becomes drier due to advection or entrainment in the early afternoon. However, given that the temperature profile in this part still shows a colder ABL, does this suggest that advection is the main reason? Ideally, free tropospheric air would be drier and warmer than the ABL air, so entrainment would dry and warm the ABL. Given the wide range of uncertainty in the figure (shaded regions), it may be worthwhile conditioning this analysis on horizontal wind speeds to characterize advection.

From your comment, we understand that you are asking about the fact that the potential temperature at the top of the ABL is cooler in the wet area than the dry area. In the response to the previous reviewer (Fig. R1), we show the individual radiosondes that make up this figure. Over all days, the wind speed throughout the ABL is low (< 5 m s-1) until the sea breeze comes (between 14-16 UTC). You are right, there are wide uncertainty bands in the radiosondes, which you can also see in Fig. R1. By selecting the composite day, we have already selected for the relatively weak wind times. That being said, even if winds are weak, gradients can be strong, which makes advection strong as well. So I would agree with you that advection is an important reason that the wet ABL is still cooler than the dry ABL (summarized with Fig. 5).

You recommend that we condition our analysis on the wind speed, and I agree that that would be a great way to separate the likely influence of advection. However, our composite day is only three days, so it is difficult to do this properly. Furthermore, advection is one of our bigger unknowns in the observations. Even in the LES, we have difficulties estimating the advection that enters the domain as models like ERA5 do not capture the mesoscale well. We have to use the LES to get a better handle on this local scale advection issue.

We have added some text to the description of Fig. 1 to address this comment and to better link the uncertainties in this figure for the need to use an LES.

> *... In the mid-afternoon, the top of the ABL is still cooler in the wet landscape than the dry landscape. This is likely due to advection, but from observations alone, we cannot quantify the advection between the wet and the dry landscapes. By the end of the afternoon, at 16 UTC, the sea-breeze arrives at the LIAISE region. The near surface atmosphere cools and moistens in both locations, but the remainder of the profile is well-mixed.*

2) Figure 3: the buoyancy flux near the surface seems smaller than the kinematic heat flux, on average. Is this just a visual issue. This needs explanation.

Thank you for pointing this out to us. We looked into it in more detail, and we found that we had an error in the calculation of buoyancy flux for the tower observations over the dry area. Other than that, the rest of Fig. 3 remains the same. For reference, we calculated buoyancy flux with:

$$\overline{w'T_v'} = \frac{1}{\rho c_p}\left[H(1 + 0.61q) + \frac{LE}{L_v}\left(0.61c_p T_a\right)\right]$$

By fixing the error in the calculation, this issue is fixed. Now, the kinematic heat flux and the buoyancy flux are nearly equivalent over the dry scale near the surface.

[Figure]

3) Line 178-179: This is a technical comment on turbulence isotropy. I suppose the authors mean component wise isotropy not turbulence isotropy, because the variances u', v' w' can be equal and turbulence remains anisotropic in the Kolmogorov sense. Also, the units of the variances are m2 s-2 not s-1

Thank you for pointing this out to us. We have fixed the description to read "component-wise isotropy" and fixed the typo in the unit in the text.

4) Line 297 and Figure 6: It is difficult to make sense of the fact that the variability in zi within a patch (irrigated vs. non irrigated) is larger than the mean difference between the patches. Can the authors show the map of surface fluxes prescribed in LES? Are these fluxes homogeneous over a patch? This may hold some clues.

The map of the surface fluxes for H and LE changes every 30 minutes (with interpolation in time by the model to prevent sudden large changes). We show a map of the Bowen ratio in Fig. 1 at 12 UTC. It is clear there that fluxes are not homogenous over the landscape and regional scales (Fig. 1 and Fig. 9). We think that these figures are sufficient to make that point. We can stress this a little bit more in the text as this is a key conclusion, but we think this is clear and has been said in the manuscript in multiple locations.

5) Line 202: The text refers to potential temperature and the figure (4) shows absolute temperature T. Which is it?

We have changed the figure to show potential temperature so that it is consistent with the rest of the work. Thank you for pointing that out to us.

6) Figure 4: Given that primes are defined as fluctuations around the full transect, shouldn't the sum of T' averaged over the wet and dry areas be zero? Similarly for uTKE. The horizontal green and yellow lines do not seem like it.

In Fig. 4, the fluctuations are defined relative to individual flight legs, but the figure shows the fluctuations averaged over multiple flight legs. The standard deviation of the fluctuations are in the shaded lines (lines 198-201 and caption of Fig. 4). Furthermore, we exclude 100 m on either side of the transition from this calculation to exclude any potential lag in the observations and errors in our method of determining the boundary. (Note, this point was not stated in the text, and we clarified this in Fig. 4's caption). While the individual $\overline{T'}$ over individual transects is zero, the $\overline{T'}_{legs,wet}$ does not necessarily equal zero.

For uTKE, we do not expect that to ever equal zero averaged over the transect as it is a sum of variances, and variances are non-zero in Reynold's averaging.

Based on this comment, we edited the caption in Fig. 4. Furthermore, we fixed the text box to more clearly indicate the variances in Fig. 4.

7) Line 385: what is meant by "counter-gradient flux" in this context? This is not clear unless the temperature and humidity profiles are themselves shown.

We added a short sentence to describe the "along-gradient" fluxes based on Fig. A1, to clarify what we mean. Therefore, the section now reads:

*For both the heat and moisture flux, there is a counter-gradient flux in the dry landscape. Based on profiles of potential temperature specific humidity from the LES (Fig. A1), the along-gradient flux in the entrainment zone is warm and dry air brought from the free atmosphere into the ABL ( $\overline{w'\theta'} < 0$ and $\overline{w'q'} > 0$). The counter-gradient fluxes could indicate the influence of the wet landscape on the ABL in the dry landscape. For TKE, there are pockets of strong TKE in the entrainment zone in the wet, although there is a skew to the higher TKE in the dry landscapes.*

8) Figure 10: stronger TKE pockets over the dry region (right part of domain) are accompanied with higher latent heat fluxes over that region, which is interesting. I don't see this discussed well, particularly in the context of a secondary circulation.

We have added some text to address this comment.

> *Fig. 10 suggests that a weak secondary circulation forms along this transect. At approximately 2 km into the dry landscape, there appears to be a small circulation that is driven by strong updrafts. In this updraft, there is high TKE coupled with a high, positive moisture flux. This shows the presence of a thermal transporting moisture that was emitted at the surface of the wet landscape towards the top of the ABL above the dry landscape. The return flow in the wet area is not as well defined in the LES. Instead we see lower ABL heights and some "dry tongues" pockets as noted by van Heerwaarden (2009) between 200 m < z < 800 m at distances -700 m and -2000 m.*

9) Figure 11: A blending height of 0.8zi (wet landscape) means that there is effectively no blending (i.e., heterogeneity effects reach far up in the ABL). Also, the difference in blending height between temperature and humidity in Fig. 11a (Alfalfa) is puzzling. Even if these have some sort of transport dissimilarity as the authors argue, such a big difference suggests that some physical mechanism is at play that was not discussed, or perhaps a consequence of the method the authors use to estimate blending height?

We agree that this result is surprising, and based on your comment, we made some changes to the section on the blending height in part to address this comment. In this response, we focus more on the differences between the temperature and moisture terms instead of specifically the case where the alfalfa blends into the wet landscape, so that we can better address this comment in the revised manuscript. We think that there are two reasons for the difference between the blending between temperature and humidity: (1) scalar dissimilarity (as mentioned in the original manuscript), and (2) humidity is more heterogeneous than potential temperature.

As we originally wrote, we hypothesized that there was scalar dissimilarity in transport between temperature and humidity. We mentioned that we hypothesized the difference in where the variables blend among the scales to be in part based on the differences between the scales. Although we generally characterized the scales from wettest to driest as alfalfa local -> wet landscape -> regional _-> dry landscape -> fallow local, the differences between each of these scales are not the same. For example, between alfalfa local and wet landscape (see the figure below), H is near zero (or negative) at the alfalfa scale, while the difference between LE is – while high – not as important for dictating the flow.

[Figure]

*Figure R5 The prescribed turbulent heat fluxes of the local and landscape scales over a day.*

The second reason for this is that the distributions of the specific humidity and moisture fluxes are more heterogeneous than those of the temperature. In the figure below, we show the probability distributions in space of all the fields in the LIAISE domain normalized by the spatial mean. We only show the surface figures, but similar results hold for the entrainment zone and the middle of the ABL. There is a higher heterogeneity in the humidity variables than the temperature variables. One part of this is the scale of the scale of the values. For example, in the mixed layer, specific humidity has a value ~10 g kg-1. Entrainment (or differences in surface fluxes) introduce variations in this of ~1 g kg-1, which is 10% of the total value. Meanwhile, if the mixed layer potential temperature has a value of ~300 K, and the deviation of it are 1-2 K, the deviations account for ~0.5% of the total value. The same holds for the fluxes. By the normalization, this feature is taken into account by our method. In fact, we consider this a feature, not a bug, of the methodology because it takes into account that small changes of humidity cause a more heterogeneous atmosphere in terms of humidity than small changes in temperature. We consider this to be one of the key conclusions of this study, so we edited the text to better describe these processes.

[Figure]

*Figure R6 The distributions of temperature (blue) and specific humidity (orange) and their turbulent fluxes at the surface normalized by the regional scale mean.*

Based on your comments, and the comments of the other reviewers, we made major changes to this section of the article in both describing the methodology (Reviewer 1) and interpreting the results (your comment).

10)Abstract Lines 9 and 10: "near 1000 m" and "~500m" are vague. Please clarify whether these represent horizontal scales or heights in the ABL.

We use "~" to indicate scales, and "near" in the meaning of "approximately", in this case referring to the actual height. We agree this is confusing terminology. We have clarified this to use ~ for scaling only and $\approx$ for approximately to indicate the (rounded) actual values of ABL height, etc.

11)Abstract Line 11: I suggest replacing "three-dimensionality of LES" with "spatiotemporal extent of LES"

Done

12) Line 47: "aggregating the land surface properties"

Done

13) Line 93: "Fig. 1 shows …"

Done

14) Line 126: replace "… temperature and humidity in a profile" with "temperature and humidity profiles".

Done

15) Line 145: "we observe …". This sentence is unclear; consider revising.

Revised

16) Line 283: remove "at"

Done

17) Line 290: Fig. 7 not 7 7

Done

18) Line 301: K/hr

Done

19) Line 303: add a period after scales.

Done

General Comments

Mangan et al. present an interesting study on atmospheric boundary layer dynamics over a heterogeneous landscape including irrigated and fallow/dry landscape units. Their main interest lies in the interactions of spatial scales and how they jointly affect atmospheric boundary layer development. In their study, they combine detailed field observations from the LIAISE campaign and Large Eddy Simulations (LES) to better understand these interactions. They use a composite day that is based on measurements during three days in July. Their approach is innovative and the detailed characterisation of boundary layer dynamics over a thermally heterogeneous landscape is unique. Overall, the manuscript is well written, and the presentation of results is clear. I think the discussion would be strengthened if the authors could discuss some of the limitations of the experimental setup. They use a composite day in the summer for their study, which allows them to discuss in detail different processes. However, how representative is this setting for other seasons/conditions? It could be discussed what would be needed to extend observations or modelling to larger timescales to investigate how representative the study conditions are.

Thank you for your comments and suggestions regarding this paper. The major suggestion is to discuss the limitations of the experimental set up and the applicability to other seasons/conditions in the discussion section. Regarding the experimental set up, the main limitation is the lack of spatial data regarding the ABL dynamics. With the radiosondes, we have two locations of hourly ABL profiles, so it is difficult to attribute features from the radiosondes into local and non-local components. Regarding the representatives of this study. Our purposes in running the "offline" (prescribed land surface fluxes) LES for a LIAISE composite day is to isolate the processes that determine the ABL, not to introduce or test ABL scaling over heterogeneous land surfaces. We want to understand how the ABL development changes based on the spatial scale. For those reasons, we do not consider this experiment to be very representative for other locations, although some of these features could still hold in other irrigated semi-arid regions, including the central valley of California and other regions in the Mediterranean. Details, like the arrival of the sea breeze and the presence of a thermal low, make it less applicable to other regions and seasons, but the irrigated crop land vs dryland contrast is more common. As this work is mainly descriptive, one should test these hypotheses in other regions and conditions.

To address these comments, we have added text and additional description to two locations in the manuscript. In Section 2.2 (LIAISE data), we added a better description of the composite day as well as a justification for doing so. The text reads:

> *We have chosen to use a composite "composite day" in this experiment to capture the inter-day variability of the atmosphere during the LIAISE experiment. The composite day is composed of three days -- 20-22 July -- which are characterized by the thermal low which developed in the north-central area of the Iberian Peninsula. During these days, synoptic conditions were similar and relatively weak. Over each of the days, there was a thermal low that developed in the Ebro River Valley, and in the late afternoons, a sea breeze bringing relatively cool and moist air from the Mediterranean Sea arrived in the study domain. Before the sea breeze, at the surface winds were primarily weak ~ 0-3 \unit{m \ s^{-1}}) from the west/northwest direction. Wind speeds reached up to 5 \unit{m \ s^{-1}} in the middle of the ABL. Furthermore, there was little daily variably in surface fluxes at each of the SEB sites over this period.*

*We calculate the composite day by averaging the available data per 30 minute period across all days. We do this both for surface fluxes and the atmospheric fields from the radiosondes, including wind speed, specific humidity and potential temperature (mean values and daily variability shown in Fig. 2). The benefits of using a composite day for analysis include (1) to gap-fill missing data, (2) reduce the spikes in the data, and (3) to create an atmospheric situation that is typical for the area and a thermal low day. Note that all processing (e.g. computing turbulent fluxes) and normalization was done on individual observations before averaging to create the composite day. This way, we create a situation that is realistic for the LIAISE experiment but not a real/case study simulation. By using a composite day approach, we can focus on the most persistent and important processes that control land-atmosphere interactions across the spatial scales.*

We have also added some text to the discussion section (in Section 5.2) to address the representativeness of the results. We briefly mention the geographical limitations as you mention here and again the potential limitations of the composite day approach. The additional text reads:

*While in this case we find that the blending height from the surface depends both on variable and spatial scale, further research should be done to investigate how generalizable this result is in both geographical location and season. We hypothesize that this multi-scaled approach is reasonable in other irrigated semi-arid regions because of typical irrigation patterns. However, as we mentioned in the formation of the LES experiments, the mesoscale forcing is vital for correctly capturing the correct mixing between the heterogeneous land surfaces. In regions with difference synoptic and mesoscale circulations, the influence of how the surface heterogeneity influences the blending in the ABL could vary.*

Specific Comments:
I do not have any major comments. However, a few sections could benefit from clarification:

Line 2: A geographical location would be useful here.

Done

Line 10: Do these results apply to afternoon conditions? If so, please specify.

Done

Figure 2: Could atmospheric boundary layer heights be added to this figure? I was also wondering if the layering could be due to the existence of a residual layer from the day before. In general, it would be good to support the interpretation of "well-mixed" with quantitative metrics. How was the mixing state determined? Did the authors apply thresholds for lapse rates?

The other reviewers made similar comments regarding marking the determined ABL height based on the observations. We have decided to add a table in an appendix to list the ABL height with time as determined from the composite radiosondes. We have also added some text to the observation section to explain why we have selected the parcel method for determining the ABL height.

We did not have a qualitive criteria for "well-mixed" instead we make this claim based on the observations (both shown in Fig. 2 and in Fig. R1). While, we agree that having a criteria for calling it well-mixed is best, we chose not to do it. We added some text in the paragraph where we describe the ABL as well mixed to qualify the claim that the ABL is well mixed.

Figure 11: The presentation of the blending height is insightful. However, I was wondering if the authors could calculate uncertainties for the blending heights. There is quite some variability, and I was not sure how much this variability is due to changing uncertainties in the estimates. Why did the authors decide to show blending heights for both scalar fluxes and concentrations? Why would the heights differ?

This is a good idea to try to quantify the uncertainties in the estimation of the blending height. To do so, we explored the thresholds of the "blending criteria". When we looked between the spatial scales, we used the somewhat arbitrary threshold of Cv of different spatial scales being within 5% of each other. As another reviewer pointed out, this criteria does not work very well when Cv approaches 0 (homogenous). In effect this induces noise because the criteria. In our case, we visually removed the noise and claim that our method won't work well above the ABL where we know that Cv approaches 0. We did some sensitivity analysis based on this criteria and found that if we use 3% or 10% as the threshold, the results of blending height do not change very much. Quantitatively, the uncertainties in this method are difficult to determine, and at the moment.

We chose to show the both the blending heights for scalar fluxes and concentrations for two reasons (1) to see if they are the same (we would expect that if w did not vary among the scales) and (2) in models that use a tiled approach to capture SGS heterogeneity, each tile has its own flux from the surface to the lowest vertical grid cell, and at that grid cell, the fluxes are aggregated. By looking at fluxes in particular, we can see if it is fair to say if there is a lowest level where the fluxes are invariant. In terms of why the humidity and temperature mixing is different, we refer to the discussion with Reviewer 2. Furthermore, we suspect that the fluxes and scalars mix at different levels because of the distribution of the thermal updrafts which in this case are important for mixing the entire ABL.

Based on your comments and the comments from the other reviewers, we have rewritten much of the section on blending height to better explain the method and cover the interpretation of the results.

**Community Comment # 1**

The authors study the diurnal variability of the atmospheric boundary layer (ABL) across spatial scales (between ~100 m and ~10 km) of irrigation-driven surface heterogeneity in the semi-arid landscape of 2021 LIAISE experiment. In my experience heterogeneity of surface fluxes in irrigated landscapes is being an emerging issue in boundary layer meteorology as we get better eyes on the system. In the old days we were satisfied to have a 100:1 fetch to height ratio and assumed the field was well watered and established a well defined surface internal boundary layer and constant flux layer. When the NCAR team went to the San Joaquin Valley to close the surface energy balance, they were in for a surprise. The adjacent fields were more or less irrigated and this caused advection

Oncley, S. P., et al. (2007), The Energy Balance Experiment EBEX-2000. Part I: overview and energy balance, *Boundary-Layer Meteorology*, *123*(1), 1-28, doi:10.1007/s10546-007-9161-1.

Now with ECOSTRESS and other high resolution sensors on the space station or satellites, we can see huge gradients in temperature across adjacent fields. So with this preamble I am excited to read this paper and see what I can learn, and the authors can teach us.

This type of work is a nice example of how we should conduct science, ID a problem in the field, make a set of field measurements to see what may be happening and then use state of are models to tie it all together, like LES in this case. Point is models may miss some processes and the measurements are never gridded as well across multiple scales as is a model. So we need each other to constrain the problem and system. And of course the answer can be conditional given season, physiological capacity and leaf area index of the crop.

Dear Dennis,

Thank you very much for the through and helpful comments on this paper, and for your positive words about the work. We apricated your enthusiasm and your good ideas on how to improve this study. Although we do not respond point-by-point to your response in this case here, we did take many ideas into account in the revised version of the manuscript. Thank you for your edit suggestions and ideas.

One of the general overarching comments you had made was about applying the results of this study to say something about local ET measurements taken with the eddy covariance tower. In particular, you mention using these results to say something about the enhance ET that has been observed over alfalfa (and other cropped) fields under these conditions. While these questions are ultimately the ones we are interested in addressing, we wanted to first understand what is going on in the ABL over the LIAISE region. For that reason, our purpose for this study is mainly to characterize the ABL and its development over the LIAISE domain. We focus on the ABL processes and how the scales of the heterogeneity mix in height and horizontally in space. In this case, we use the LES with prescribed surface fluxes, so there is no feedback on ABL processes on ET. Because we are prescribing the surface fluxes, we incorporate the observed enhanced evaporation to the surface fluxes, but we cannot say directly from this case why there is enhanced evaporation in the first place.

Currently, we are working on a similar LES case for LIAISE with a coupled land surface model. This way we can take into account the atmospheric feedbacks (via advection and entrainment) on the ET. We particularly focus on the ET over the alfalfa field so that we can better understand the observed flux profiles in the surface layer. We will reach out to you directly to discuss this work.

Sincerely,

Mary Rose Mangan

**References**

Angevine, W. M., Baltink, H. K., & Bosveld, F. C. (2001). Observations Of The Morning Transition Of The Convective Boundary Layer. *Boundary-Layer Meteorology*, *101*(2), 209–227. https://doi.org/10.1023/A:1019264716195

Barrett, A. I., Hogan, R. J., & O'Connor, E. J. (2009). Evaluating forecasts of the evolution of the cloudy boundary layer using diurnal composites of radar and lidar observations. *Geophysical Research Letters*, *36*(17). https://doi.org/10.1029/2009GL038919

Grotjahn, R., & Faure, G. (2008). *Composite Predictor Maps of Extraordinary Weather Events in the Sacramento, California, Region*. https://doi.org/10.1175/2007WAF2006055.1

Jiménez, M. A., Grau, A., Martínez-Villagrasa, D., & Cuxart, J. (2023). Characterization of the marine-air intrusion Marinada in the eastern Ebro sub-basin. *International Journal of Climatology*, *43*(16), 7682–7699. https://doi.org/10.1002/joc.8287

Jury, M. R., Chiao, S., & Harmsen, E. W. (2009). Mesoscale Structure of Trade Wind Convection over Puerto Rico: Composite Observations and Numerical Simulation. *Boundary-Layer Meteorology*, *132*(2), 289–313. https://doi.org/10.1007/s10546-009-9393-3

Lunel, T., Boone, A. A., & Le Moigne, P. (2024). Irrigation strongly influences near-surface conditions and induces breeze circulation: Observational and model-based evidence. *Quarterly Journal of the Royal Meteorological Society*. https://doi.org/10.1002/qj.4736

Mangan, M. R., Hartogensis, O., Boone, A., Branch, O., Canut, G., Cuxart, J., de Boer, H. J., Le Page, M., Martínez-Villagrasa, D., Miró, J. R., Price, J., & Vilà-Guerau de Arellano, J. (2023). The surface-boundary layer connection across spatial scales of irrigation-driven thermal heterogeneity: An integrated data and modeling study of the LIAISE field campaign. *Agricultural and Forest Meteorology*, *335*, 109452. https://doi.org/10.1016/j.agrformet.2023.109452

Mangan, M. R., Hartogensis, O., van Heerwaarden, C., & Vilà-Guerau de Arellano, J. (2023). Evapotranspiration controls across spatial scales of heterogeneity. *Quarterly Journal of the Royal Meteorological Society*, *149*(756), 2696–2718. https://doi.org/10.1002/qj.4527

Maronga, B., & Raasch, S. (2013). Large-Eddy Simulations of Surface Heterogeneity Effects on the Convective Boundary Layer During the LITFASS-2003 Experiment. *Boundary-Layer Meteorology*, *146*(1), 17–44. https://doi.org/10.1007/s10546-012-9748-z

May, P. T., & Wilczak, J. M. (1993). *Diurnal and Seasonal Variations of Boundary-Layer Structure Observed with a Radar Wind Profiler and RASS*. https://journals.ametsoc.org/view/journals/mwre/121/3/1520-0493_1993_121_0673_dasvob_2_0_co_2.xml

Otte, M. J., & Wyngaard, J. C. (2001). *Stably Stratified Interfacial-Layer Turbulence from Large-Eddy Simulation*. https://journals.ametsoc.org/view/journals/atsc/58/22/1520-0469_2001_058_3424_ssiltf_2.0.co_2.xml

Paleri, S., Wanner, L., Sühring, M., Desai, A., & Mauder, M. (2023). Coupled large eddy simulations of land surface heterogeneity effects and diurnal evolution of late summer and early autumn atmospheric boundary layers during the CHEESEHEAD19 field campaign. *EGUsphere*, 1–43. https://doi.org/10.5194/egusphere-2023-1721

Paleri, S., Wanner, L., Sühring, M., Desai, A. R., Mauder, M., & Metzger, S. (2024). Impact of Surface Heterogeneity Induced Secondary Circulations on the Atmospheric Boundary Layer. *Boundary-Layer Meteorology*, *191*(1), 3. https://doi.org/10.1007/s10546-024-00893-7

Patton, E. G., Sullivan, P. P., & Moeng, C.-H. (2005). The Influence of Idealized Heterogeneity on Wet and Dry Planetary Boundary Layers Coupled to the Land Surface. *Journal of the Atmospheric Sciences*, *62*(7), 2078–2097. https://doi.org/10.1175/JAS3465.1

Philibert, A., Lothon, M., Amestoy, J., Meslin, P.-Y., Derrien, S., Bezombes, Y., Campistron, B., Lohou, F., Vial, A., Canut-Rocafort, G., Reuder, J., & Brooke, J. K. (2024). CALOTRITON: A convective boundary layer height estimation algorithm from ultra-high-frequency (UHF) wind profiler data. *Atmospheric Measurement Techniques*, *17*(6), 1679–1701. https://doi.org/10.5194/amt-17-1679-2024

Raasch, S., & Harbusch, G. (2001). An Analysis Of Secondary Circulations And Their Effects Caused By Small-Scale Surface Inhomogeneities Using Large-Eddy Simulation. *Boundary-Layer Meteorology*, *101*(1), 31–59. https://doi.org/10.1023/A:1019297504109

Schalkwijk, J., Jonker, H. J. J., Siebesma, A. P., & Bosveld, F. C. (2015). A Year-Long Large-Eddy Simulation of the Weather over Cabauw: An Overview. *Monthly Weather Review*, *143*(3), 828–844. https://doi.org/10.1175/MWR-D-14-00293.1

Segal, M., & Arritt, R. W. (1992). *Nonclassical Mesoscale Circulations Caused by Surface Sensible Heat-Flux Gradients*. https://journals.ametsoc.org/view/journals/bams/73/10/1520-0477_1992_073_1593_nmccbs_2_0_co_2.xml

Skamarock, W. C. (2004). *Evaluating Mesoscale NWP Models Using Kinetic Energy Spectra*. https://doi.org/10.1175/MWR2830.1

van Heerwaarden, C. C., Mellado, J. P., & Lozar, A. D. (2014). Scaling Laws for the Heterogeneously Heated Free Convective Boundary Layer. *Journal of Atmospheric Sciences*, *71*(11), 3975–4000. https://doi.org/10.1175/JAS-D-13-0383.1

van Stratum, B. J. H., van Heerwaarden, C. C., & Vilà-Guerau de Arellano, J. (2023). The Benefits and Challenges of Downscaling a Global Reanalysis With Doubly-Periodic Large-Eddy Simulations. *Journal of Advances in Modeling Earth Systems*, *15*(10), e2023MS003750. https://doi.org/10.1029/2023MS003750